# Contrastive Unlearning: A Contrastive Approach to Machine Unlearning

## Abstract

Machine unlearning aims to eliminate the influence of a subset of training samples (i.e., unlearning samples) from a trained model. Effectively and efficiently removing the unlearning samples without negatively impacting the overall model performance is challenging. Existing works mainly exploit input and output space and classification loss, which can result in ineffective unlearning or performance loss. In addition, they utilize unlearning or remaining samples ineffectively, sacrificing either unlearning efficacy or efficiency. Our main insight is that direct optimization on the representation space utilizing both unlearning and remaining samples can effectively remove influence of unlearning samples while maintaining representations learned from remaining samples. We propose a contrastive unlearning framework, leveraging the concept of representation learning for more effective unlearning. It removes the influence of unlearning samples by contrasting their embeddings against the remaining samples' embeddings so that their embeddings are closer to the embeddings of unseen samples. Experiments on a variety of datasets and models on both class unlearning and sample unlearning showed that contrastive unlearning achieves the best unlearning effects and efficiency with the lowest performance loss compared with the state-of-the-art algorithms.

## 1 Introduction

Machine unlearning Cao & Yang (2015) aims to remove a subset of data (i.e., unlearning samples) from a trained machine learning (ML) model without retraining the model from scratch and has received increasing attention due to various privacy regulations. Notably, "the right to be forgotten" from the General Data Protection Requirement (GDPR) gives individuals the right to request their data to be removed from databases, which extends to models trained on such data (Mantelero, 2024). Since models can remember training data within their parameters Arpit et al. (2017), it is necessary to "unlearn" these data from a trained model. The goals and evaluation metrics for unlearning typically include: 1) unlearning efficacy, which measures how well the algorithm removes the influence of unlearning samples. This can be assessed by the model's performance on the unlearning samples, or by its robustness against membership inference attacks Shokri et al. (2017) using unlearning samples; 2) model performance on its original tasks, which ensures that the unlearning does not significantly degrade its overall accuracy; and 3) computational efficiency, which assesses the time and resources required for the unlearning.

While many promising approaches are proposed, existing works present several limitations: 1) they mainly exploit input and output space and classification loss. It produces significant shift in decision boundaries. 2) They either focus on unlearning or remaining samples alone or use both but in an ineffective way and hence either sacrifice the unlearning efficacy or efficiency. For example, Gradient Ascent Golatkar et al. (2020) only uses unlearning samples and attempts to reverse their impact by applying gradient *ascent* using the classification loss. Finetune Golatkar et al. (2020) only uses remaining samples to iteratively retrain the model to gradually remove the influence of unlearning samples leveraging the catastrophic forgetting effect (Goodfellow et al., 2013). SCRUB Kurmanji et al. (2023) uses both unlearning and remaining samples for unlearning, but requires multiple iterations over the entire remaining samples, leading to excessive computations.

**Our Contributions.** To address these deficiencies, we present a novel contrastive approach for machine unlearning, or **contrastive unlearning**. We rethink the problem of machine unlearning

in the perspective of representation space. We re-purpose the idea of supervised contrastive learning Khosla et al. (2020), a widely used representation learning approach, for more effective unlearning. Optimizing representation space is more effective because it allows direct adjustments of unlearning samples without excessive transformation of decision boundaries. Simultaneously, it is more efficient since it only optimizes embeddings of unlearning samples and small portion of remaining samples.

A fully trained model perceives training and test samples differently. When test samples are given to the model, most of their embeddings land within the correct decision boundary. However, since the model was not optimized against the test (unseen) samples, their embeddings are located closer to the decision boundary than those of the training samples. If the embeddings of the unlearning samples become indistinguishable from the embeddings of the test samples, we can claim that the model is no longer influenced by the unlearning samples. Thus, the goal of unlearning is to adjust the model so it produces embeddings of the unlearning samples similar to the embeddings of the test samples.

Based on the idea, given an unlearning sample, we contrast it with 1) Positive samples (remaining samples from the same class as the unlearning sample) and push their embeddings apart from each other, and 2) Negative samples (remaining samples from different classes as the unlearning sample) and pull their embeddings close to each other. This results in the unlearned embedding to be geometrically distant from remaining samples and closer to the decision boundaries and test samples' embeddings. It has two main insights. First, directly optimizing the embeddings of unlearning samples, which captures the most important features of the samples being memorized, facilitates more effective unlearning. Second, by contrasting unlearning and subset of remaining samples during unlearning and using both positive and negative remaining samples as references for optimizing the embedding of unlearning samples, it can effectively remove the influence of unlearning samples while minimizing any change of the decision boundaries of remaining samples. Additionally we introduce an auxiliary classification loss on the contrasted remaining samples to further maintain model accuracy.

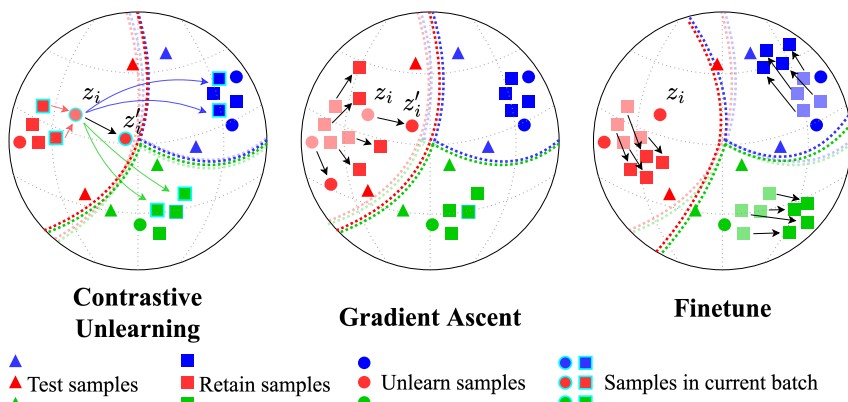

Figure 1: Visualization of Representation Spaces for Unlearning, Gradient Ascent, and Fine-Tuning

Figure 1 illustrates the intuition of contrastive unlearning in comparison to existing approaches in a normalized representation space. Circles and squares are embeddings of the unlearning samples and remaining samples. Triangles are embeddings of test samples. Colors represent different classes. Dotted lines show decision boundaries. We assume the model has been trained, so that the embeddings of training samples are clustered to their respective classes Das & Chaudhuri (2024).

Given an embedding of unlearning sample $z_i$, contrastive unlearning pushes $z_i$ away from its own class (positive pairs) and pulls $z_i$ towards the samples with different classes (negative pairs). This results in the unlearned embedding $z_i'$ to be distant from remaining samples and closer to the decision boundaries, where test samples' embeddings (triangles) are located. In comparison, Gradient ascent Golatkar et al. (2020) attempts to reverse the impact of unlearning samples. It pushes $z_i$ away in the representation space but is difficult to obtain a proper unlearn efficacy and performance. It either applies insufficient change in the decision boundary of classes (ineffective unlearning), or it may significantly affect embeddings of remaining samples of the same class (model utility loss).

Finetune attempts to train the model only using remaining samples. In representation space, this only indirectly pushes the unlearning samples away (ineffective unlearning) and is susceptible to overfitting to the remaining samples (model utility loss).

Our contrastive *unlearning* is fundamentally different from contrastive learning since the goal of contrastive learning is to distinguish different samples, while our goal is to modify embeddings of particular unlearning samples and maintain model's general classification performance. It features several novel algorithm designs and new findings: 1) we construct contrasting pairs different from conventional contrastive learning to serve the unlearning purpose and design new contrastive unlearning losses for both sample unlearning (unlearning randomly selected training samples) and single class unlearning (unlearning every sample of a class) tasks; 2) while it is common to add a classification loss to maintain the performance of the unlearning model, through the new lens of contrastive unlearning, we find that the classification loss helps keep the embeddings of the remaining samples in place and reciprocally improves unlearning effectiveness, validated by our empirical analysis followed by in-depth analysis. In addition, contrastive unlearning is highly scalable as it can be implemented on top of various contrastive learning algorithms. While our analysis is based on supervised contrastive learning Khosla et al. (2020), we show that contrastive unlearning can be implemented with Momentum Contrast (MoCo) He et al. (2020). Also, contrastive unlearning is not restricted to unlearning classification models. We show that it is capable of unlearning other models such as vision-language models trained with contrastive loss.

We conduct comprehensive experiments on both class unlearning and sample unlearning to demonstrate the effectiveness and versatility of our approach in comparison to state-of-the-art methods. Experimental results show that contrastive unlearning achieves the most effective unlearning (low model accuracy on unlearning samples comparable to the retrained model) while maintaining model utility (high model accuracy on test samples), with high computation efficiency. In addition, we conduct a membership inference attack (MIA) Shokri et al. (2017) for deeper verification of unlearning. We assume a strong adversary who has full access to the unlearned model, simulating an administrator who conducted unlearning and wants to verify the effectiveness of unlearning (Thudi et al., 2022; Cotogni et al., 2023). Contrastive unlearning has the lowest member prediction rate on unlearning samples compared to all baselines, indicating the most effective unlearning. To enhance scalability of our model, we show experimental results of contrastive unlearning based on MoCo He et al. (2020). Also we show the versatility and generalizability of contrastive unlearning by providing the results of removing a class from a few-shot image-language classifier Radford et al. (2021).

In summary, our contributions are as follows:

(1) We propose contrastive unlearning, an algorithm utilizing the concept of contrastive loss. We achieve unlearning by modifying embeddings of unlearning samples to be similar to the embeddings of test samples (unseen samples) without directly using them. With a contrastive approach, we effectively and efficiently remove the influence of unlearning samples by adjusting their embeddings.

(2) We design a contrastive unlearning loss that effectively captures and removes the most important features relevant for classification from the embeddings of the unlearning samples (achieving effective unlearning) while keeping the embeddings of remaining samples intact (maintaining model utility). We design a contrastive loss for two tasks: single class unlearning and random sample unlearning.

(3) We conduct comprehensive experiments comparing contrastive unlearning with various state-of-the-art methods on two unlearning tasks, single class and sample unlearning, to demonstrate the effectiveness and versatility of our approach. We also conduct a membership inference attack to verify the unlearning efficacy. The results show that contrastive unlearning has the best efficacy while maintaining model utility with high computational efficiency.

## 2 RELATED WORKS

Machine unlearning was introduced by Cao & Yang (2015) with two goals: completeness, suggesting an unlearning algorithm should reverse the influence of unlearning samples and the unlearned model should be consistent with a model retrained only from remaining samples; and timeliness, requiring the running time of the unlearning algorithm to be faster than retraining. The unlearned

model should maintain high performance after unlearning. Exact unlearning ensures the completeness of unlearning. SISA is an exact unlearning framework that splits the dataset into partitions and trains a model for each shard. Given an unlearning request, it retrains models whose shard has the unlearning sample (Bourtoule et al., 2021). ARCANE uses a partitioning strategy by sample classes (Yan et al., 2022). These frameworks require partitioned training and still expensive retraining computation, and model performance is highly dependent on partitioning strategy (Koch & Soll, 2023).

Approximate unlearning allows approximate completeness. Certified unlearning provides a mathematical guarantee on the approximation. Guo et al. (2020) proposed unlearning using newton-type hessian update with $(\varepsilon, \delta)$-indistinguishability. Neel et al. (2024) proposes an algorithm based on project gradient descent on the partitioned dataset with a probabilistic bound. Approximation guarantee is also useful for graph unlearning (Wu et al., 2023; Zhang, 2024). Gupta et al. (2021) further studied correlation of unlearning requests proposed adaptive unlearning streams. Fisher unlearning uses Fisher information matrix Golatkar et al. (2020) to identify optimal noise to remove the influence of unlearning samples. Drawbacks on certified unlearning algorithms are the difficulty to scale, and most of them requires convexity is required for the mathematical guarantee. Moreover, Thudi *et* al. Thudi et al. (2022), questioned validity of certified unlearning. Recently, some works tried to resolve limitations of certified unlearning. Metha *et* al proposed LCODEC Mehta et al. (2022), which reduced the computation cost by selectively generating Hessian matrices. Also, Zhang *et* al. proposed certified unlearning for non-convex setting (Zhang et al., 2024). While both are promising, however, their experimental results show suboptimal unlearn efficacy.

Another body of approximate unlearning shows the unlearning effect through empirical evaluations. Usually, these works target class unlearning, which is to unlearn every sample of a class. UNSIR Tarun et al. (2023) conducts noisy gradient updates using the unlearning class. Boundary unlearning unlearns an entire class Chen et al. (2023) by changing decision boundaries. ERM-KTP uses a special neural architecture known as an entanglement reduce mask (Lin et al., 2023). SCRUB Kurmanji et al. (2023) is based on the teacher-student network, where the teacher or the original model transfers knowledge to the unlearned model in every class except the unlearning class. Recently, Cha *et* al. proposed an instance-wise unlearning using cross-entropy loss Cha et al. (2024). Similar to our work, the authors provided analysis on decision boundaries. Our approach is an approximate unlearning method for both sample and class unlearning. We compare it with both types of methods, as well as empirical and certified methods, showing its superiority through empirical evaluations. We do not compare Cha et al. (2024) as its assumptions and goal of unlearning does not align with our problem settings. The authors assume that remaining samples are unavailable, and their unlearning goal of unlearning is to incorrectly classify all unlearning samples. However, we assume that remaining samples are available and our goal of unlearning is to make the model to perceive unlearning samples as unseen samples.

## 3 PROBLEM DEFINITION

We define a classification model $\mathcal{F} = \mathcal{H}(\mathcal{E}_\theta(\cdot))$ where $\mathcal{E}_\theta(\cdot)$ is a neural network based encoder parameterized by $\theta$ and $\mathcal{H}(\cdot)$ is a classification head. $\mathcal{E}_\theta$ produces embeddings $z$ given a sample $x$. $\mathcal{H}$ receives $z$ and yields a prediction. Let $\mathcal{F}$ be trained using dataset $\mathcal{D}_{tr} = \{(x_1, y_1) \cdots (x_n, y_n)\}$, where each data point is a tuple $(x_i, y_i)$ including feature set $x_i$ and label $y_i \in \{0 \cdots C\}$ where $C$ is the number of classes. We suppose $\mathcal{F}$ was trained with cross-entropy loss. Let $\mathcal{D}_{ts}$ be a test dataset sampled from an analogous distribution with $\mathcal{D}_{tr}$, satisfying $\mathcal{D}_{ts} \cap \mathcal{D}_{tr} = \emptyset$.

Let $\mathcal{D}_{tr}^u \subseteq \mathcal{D}_{tr}$ be a set of samples to be forgotten (i.e., unlearning samples). The remaining set is $\mathcal{D}_{tr}^r = \mathcal{D}_{tr} \setminus \mathcal{D}_{tr}^u$. Let a retrained model $\mathcal{F}^R$ be trained only with $\mathcal{D}_{tr}^r$. An unlearning algorithm $M$ receives $\mathcal{D}_{tr}^r, \mathcal{D}_{tr}^u, \theta$ and produces $\theta'$. An unlearned model $\mathcal{F}' = \mathcal{H}(\mathcal{E}_{\theta'})$ should resemble $\mathcal{F}^R$.

### 3.1 SINGLE CLASS UNLEARNING

For single class unlearning, $D_{tr}^u$ consists of all samples of an unlearning class $c$. The test set $\mathcal{D}_{ts}$ can be split into $\mathcal{D}_{ts}^u$ and $\mathcal{D}_{ts}^r$, where $\mathcal{D}_{ts}^u$ includes all test samples of class $c$, and $\mathcal{D}_{ts}^r = \mathcal{D}_{ts} \setminus \mathcal{D}_{ts}^u$ includes all test samples of remaining classes. A retrained model $\mathcal{F}^R$ will have zero accuracy on $\mathcal{D}_{tr}^u$ and $\mathcal{D}_{ts}^u$, the training and test samples of class $c$, since it was retrained without class $c$. So given an

accuracy function $Acc$, the goal of single class unlearning is for the unlearned model $\mathcal{F}'$ to achieve near-zero accuracy on both training and test samples of class $c$ and similar accuracy as the retrained model $\mathcal{F}^R$ for remaining classes.

$$\text{Acc}\left(\mathcal{F}', \mathcal{D}_{tr}^u\right) \approx 0, \quad \text{Acc}\left(\mathcal{F}', \mathcal{D}_{ts}^u\right) \approx 0, \tag{1}$$

$$\text{Acc}\left(\mathcal{F}', \mathcal{D}_{ts}^r\right) \approx Acc\left(\mathcal{F}^R, \mathcal{D}_{ts}^r\right). \tag{2}$$

Single-class unlearning can be implemented using simple rules. For example, the rule can assign random labels to samples classified as target class. However, rule-based unlearning has significant limitations for the following reasons: (1) Insufficient Unlearning: Learned patterns of samples from the unlearning class remain embedded within the model's weights. If the model's weights are leaked, an adversary can potentially recover knowledge of the unlearning class. (2) Model Utility: Rule-based unlearning can potentially degrade the performance of all remaining classes.

### 3.2 SAMPLE UNLEARNING

For sample unlearning, the unlearning samples $\mathcal{D}_{tr}^u$ can belong to different classes. A retrained model $\mathcal{F}^R$ will have similar accuracy on unlearning samples $\mathcal{D}_{tr}^u$ and test samples $\mathcal{D}_{ts}$ since unlearning samples are not in the training set anymore. So the goal of sample unlearning is for the unlearned model $\mathcal{F}'$ to achieve similar accuracy as the retrained model $\mathcal{F}^R$ on both unlearning samples and test samples.

$$\text{Acc}\left(\mathcal{F}', \mathcal{D}_{tr}^u\right) \approx Acc\left(\mathcal{F}^R, \mathcal{D}_{ts}\right), \tag{3}$$

$$\text{Acc}\left(\mathcal{F}', \mathcal{D}_{ts}\right) \approx Acc\left(\mathcal{F}^R, \mathcal{D}_{ts}\right). \tag{4}$$

A more generalized model can easily achieve sample unlearning as it can easily achieve Equation 3. While it can achieve certain level of unlearning, we deem that generalization is not sufficient as it eventually allows model to obtain unique pattern of training samples Long et al. (2018).

## 4 CONTRASTIVE UNLEARNING

Contrastive unlearning utilizes geometric properties of representation space for unlearning purposes and leverages the contrast between remaining and unlearning samples. If a sample $x$ had been used as a training example, information extracted from $x$ by $\mathcal{E}_\theta$ would be geometrically expressed in the representation space. Specifically, we hypothesize that samples of a same class have similar embeddings and samples from different classes have dissimilar embeddings even when the model was not explicitly trained with representation learning. This can be supported by existing literature, which mathematically and empirically showed that a model optimized with cross-entropy loss produces higher geometric similarity among embeddings of samples of the same class and lower similarity among different classes (Das & Chaudhuri, 2024; Graf et al., 2021).

From this intuition, we modify characteristics of representation space of unlearning samples to be similar to the representation of unseen samples. We aim to isolate embeddings of unlearning samples away from remaining samples up to the point where the model perceives them as unseen samples. To effectively achieve this, we contrast each unlearning sample with 1) remaining samples from the same class (positive pairs) and push their representations apart from each other, and 2) remaining samples from different classes (negative pairs) and pull their representations close to each other. To this end, the embeddings of unlearning samples approach to the decision boundaries of the classes. This has some relation with existing literature of contrastive learning, however, our approach is fundamentally different as it contrasts pairs of unlearning and remaining samples while contrastive learning contrasts samples simply by their classes.

**Contrastive Unlearning Loss: Sample Unlearning.** Contrastive unlearning uses a batched process. In each round, an unlearning batch $X^u = \{x_1^u, \cdots x_B^u\}$ with size $B$ is sampled from the unlearning data $\mathcal{D}_{tr}^u$, and a remaining batch $X^r = \{x_1^r \cdots x_B^r\}$ is sampled from the remaining set

$\mathcal{D}_{tr}^r$. We denote $x_i$ as $i$-th sample of $X^u$ as an anchor. Based on the anchor $x_i$, positives and negatives are chosen from $X^r$. Positives are $P_{\mathbf{x}}(x_i) = \{x_j | x_j \in X^r, y_j = y_i\}$, or remaining samples with the same class as $x_i$; negatives are $N_{\mathbf{x}}(x_i) = \{x_j | x_j \in X^r, y_j \neq y_i\}$, or remaining samples with different class as $x_i$. Correspondingly, let embeddings of positives and negatives be $P_{\mathbf{z}}(x_i) = \{z_j | z_j = \mathcal{E}_\theta(x_j), x_j \in P_{\mathbf{x}}(x_i)\}$ and $N_{\mathbf{z}}(x_i) = \{z_j | z_j = \mathcal{E}_\theta(x_j), x_j \in N_{\mathbf{x}}(x_i)\}$. The contrastive unlearning loss aims to minimize the similarity of positive pairs and maximizes the similarity of negative pairs (the opposite of contrastive learning).

$$\mathcal{L}_{UL} = \sum_{x_i \in X^u} \frac{-1}{|N_{\mathbf{z}}(x_i)|} \sum_{z_a \in N_z} \log \frac{\exp(z_i \cdot z_a / \tau)}{\sum_{z_p \in P_{\mathbf{z}}(x_i)} \exp(z_i \cdot z_p / \tau)} \tag{5}$$

where $\tau \in \mathcal{R}^+$ is a scalar temperature parameter. In our final algorithm, we contrast each $X^u$, with $\omega$ randomly sampled $X^r$. Thus within a single unlearning round, our algorithm computes every batch of $\mathcal{D}_{tr}^u$ for $\omega$ times. Refer to appendix B for more details.

**Contrastive Unlearning Loss: Single Class Unlearning.** For single class unlearning, the unlearning set $\mathcal{D}_{tr}^u = \{(x_i, y_i) | y_i = c\}$ and remaining set $\mathcal{D}_{tr}^r = \{(x_i, y_i) | y_i \neq c\}$. This makes the positive set $P_{\mathbf{z}} = \emptyset$ as none of remaining samples belong to class $c$. In short, there are no positive remaining samples to push away the unlearning samples. Thus we change equation 5 as follows.

$$\mathcal{L}_{UL} = \sum_{x_i \in X^u} \frac{-1}{|N_{\mathbf{z}}(x_i)|} \sum_{z_a \in N_z} \log \frac{\exp(z_i \cdot z_a / \tau)}{|N_z(x_i)|}. \tag{6}$$

We replaced the previous denominator to $|N_{\mathbf{z}}(x_i)|$. This is because equation 5 requires both directions to push and pull unlearning samples. Lacking one of the directions increases the instability of the loss. Since $P_z = \emptyset$, we replace the denominator to $|N_{\mathbf{z}}(x_i)|$ to introduce damping effects against excessively pulling unlearning samples to negative samples.

**Classification Loss of Remaining Samples.** A novel challenge of contrastive unlearning is to preserve embeddings of remaining samples. Optimizing equation 5 not only alters embeddings of the anchor unlearning sample but also reciprocally alters embeddings of all samples in $P_{\mathbf{x}}$ and $N_{\mathbf{x}}$. All positive samples are slightly pushed away from and all negatives are slightly pulled toward the anchor. A similar effect arises in contrastive learning, but it is not problematic as it reinforces the consolidation of embeddings of the same class, which is a desired effect. However, for unlearning purposes, embeddings of $X^r$ have to be preserved, because: 1) not preserving them directly leads to a loss in model performance, and 2) it also reciprocally affects unlearning effectiveness as magnitude of pulling and pushing decreases. In short, embeddings of $X^r$ are also modified as a byproduct of optimization and it is necessary to restore them back. We utilize cross-entropy loss for restoring embeddings of $X^r$, because it derives maximum likelihood independently to each sample Shore & Johnson (1981). This ensures obtaining directions very close to the original embeddings no matter how embeddings of remaining samples are modified. Combining the unlearning loss, the final loss for our proposed contrastive unlearning is as follows,

$$\mathcal{L} = \lambda_{UL} \mathcal{L}_{UL} + \lambda_{CE} \mathcal{L}_{CE}(\mathcal{F}(X^r), Y^r), \tag{7}$$

where $X^r$ and $Y^r$ are batched remaining samples and their corresponding labels. $\lambda_{CE}$ and $\lambda_{UL}$ are hyperparamters to determine influence of two loss terms. The full algorithm is in the appendix B.

**Termination Condition.** The termination condition for the algorithm differs based on the task of unlearning. We assume a small dataset $\mathcal{D}_{\text{eval}}$ is available for evaluation. The algorithm evaluates $\mathcal{F}'$ with $\mathcal{D}_{eval}$ and terminates if it satisfies unlearning criteria. For single class unlearning, $\mathcal{D}_{\text{eval}} = D_{ts}^u$, the test data of the unlearning class. The algorithm terminates when the accuracy of the unlearned model $\mathcal{F}'$ on the unlearning class falls below a threshold where $C$ is the total number of classes in the training data and $1/C$ corresponds to the accuracy of a random guess.

$$\text{Acc}(\mathcal{F}', \mathcal{D}_{\text{eval}}) \leq \frac{1}{C}. \tag{8}$$

For sample unlearning, $\mathcal{D}_{\text{eval}} = \{\mathcal{D}_{eval}^u, \mathcal{D}_{eval}^{ts}\}$ where $\mathcal{D}_{eval}^u \subseteq \mathcal{D}_{tr}^u$ and $\mathcal{D}_{eval}^{ts} \subseteq \mathcal{D}_{ts}$. The algorithm terminates when the accuracy of $\mathcal{F}'$ on the unlearning samples $\mathcal{D}_{eval}^u$ drops below the accuray on test samples $\mathcal{D}_{eval}^{ts}$.

$$\text{Acc}(\mathcal{F}', \mathcal{D}_{eval}^u) \leq \text{Acc}(\mathcal{F}', \mathcal{D}_{eval}^{ts}). \tag{9}$$

The termination conditions are proxy conditions that loosely satisfies problem definition of 3.1 and 3.2. In single class unlearning, retrained model provides zero accuracy on unlearning class. An

unlearned model should behave identically on unlearning class. However, it can be challenging for unlearning algorithms to achieve the zero accuracy. Thus we loose the condition and consider sufficient amount of knowledge is removed once the model satisfies the inequality 8 (corresponding to a random guess). In sample unlearning, it is not desired to terminate the algorithm before satisfying the condition in 9 because it implies that the model still retains information regarding $\mathcal{D}_{tr}^u$. It is also not desired to continue running the algorithm to further reduce accuracy on $\mathcal{D}_{tr}^u$ much lower than $\mathcal{D}_{ts}$ because it does not align with definition of sample unlearning from section 3.2 as it is negatively injecting information regarding $\mathcal{D}_{tr}^u$ into $\theta'$. This results in $\mathcal{F}'$ to deliberately make incorrect classification on $\mathcal{D}_{tr}^u$, which is not aligned with the goal of sample unlearning.

## 5 EXPERIMENTS

### 5.1 EXPERIMENT SETUP

**Datasets and Models.** We use three benchmark datasets: CIFAR-10, SVHN, and Mini-Imagenet Cao (2022), and employ ResNet(RN)-18, 34, 50, and 101 models He et al. (2016) and ViT-small Dosovitskiy et al. (2021) in our experiments. Refer to the appendix for details on the original models, implementations (code), SVHN and Mini-Imagenet experiments, unlearning few-shot CLIP model Radford et al. (2021) and unlearning based on MOCO He et al. (2020) .

**Comparison Methods.** For class unlearning, we remove all samples belonging to class 5 by default. For sample unlearning, we remove randomly selected 500 samples by default. We also evaluate class unlearning on other classes and sample unlearning of varying number of samples. Please refer to the appendix for results. To assure the robustness, we repeat sample unlearning with a random seed for five times and report the average and standard deviation of the results. For both tasks, we provide **Retrain**, a retrained model using the training data excluding the unlearning class or samples, as an ideal reference for unlearning efficacy and model performance.

We include four state-of-the-art methods specifically designed for **single class unlearning**: 1) **Boundary Expansion** Chen et al. (2023) trains the model using all unlearning samples as a temporary class and then discards the temporary class. 2) **Boundary Shrink** Chen et al. (2023) is similar to Boundary Expansion but it modifies the decision boundary of unlearning class to prevent unlearning samples from being classified into the unlearning class (unlearning samples are classified as other classes). 3) **SCRUB** Kurmanji et al. (2023) is based on the teacher-student framework and selectively transfers information from the original model to the unlearned model (all information except that of the unlearning class). 4) **UNSIR** Tarun et al. (2023) uses an iterative process of impairing and recovering and generates noise that maximizes error in the unlearning class and repairs the classification performance for the other classes.

We include four state-of-the-art methods designed for **sample unlearning**: 1) **Finetune** Golatkar et al. (2020) leverages catastrophic forgetting Goodfellow et al. (2013) and iteratively trains the original model only using the remaining samples. 2) **Gradient Ascent** Golatkar et al. (2020) conducts gradient ascent using unlearning samples. 3) **Fisher** Golatkar et al. (2020) is a certified unlearning algorithm using randomization techniques borrowed from differential privacy and leverages the Fisher information matrix to design optimal noise for noisy gradient updates. 4) **LCODEC** Mehta et al. (2022) is also a certified unlearning method that proposes a fast and effective way of obtaining Hessian by selecting parameters by their importance.

We note that sample unlearning methods may be used for class unlearning. However, our class unlearning baselines already demonstrated their superiority over the sample unlearning baeslines including Finetune, Gradient Ascent, and Fisher, hence we do not include them in comparison.

**Evaluation Metrics. Model performance** is assessed by accuracy of the unlearned model on the test data of remaining classes $\mathcal{D}_{ts}^r$ (class unlearning) and on the test data $\mathcal{D}_{ts}$ (sample unlearning). The accuracy should be similar to the retrained model. **Unlearning efficacy** is assessed by accuracy of the unlearned model on the training and test data of unlearning class $\mathcal{D}_{tr}^u$ and $\mathcal{D}_{ts}^u$ (class unlearning) and the unlearning samples $\mathcal{D}_{tr}^u$ (sample unlearning). A successful class unlearning should achieve zero accuracy on train and test data of unlearning class. For sample unlearning, we provide an additional metric of unlearn score, which is the absolute difference between the accuracy of test samples and unlearn samples. A successful sample unlearning should achieve a low unlearn score

which means the model perceives unlearning samples and test samples (unseen samples) similarly. **Efficiency** is measured by the runtime of the unlearning algorithm. A shorter runtime indicates better efficiency.

**Unlearning Verification via MIA.** We conduct a membership inference attack (MIA) Shokri et al. (2017) to verify sample unlearning. We assume an adversary with full access to the unlearned model and training data, simulating an administrator who conducted unlearning and uses MIA to verify the effectiveness of unlearning (Thudi et al., 2022; Cotogni et al., 2023). Although more robust MIA frameworks are available such as LiRA Carlini et al. (2022), we used the MIA framework from Shokri et al. (2017) as our main goal is to fairly compare our contrastive unlearning and other baseline unlearning algorithms and to obtain a generalizable comparison on unlearn efficacy.

To train the attack model, we sample $\mathcal{D}^M$ from remaining samples $\mathcal{D}^r_{tr}$ (as members) and $\mathcal{D}^N$ from test samples $\mathcal{D}_{ts}$ (as non-members). An attack model is trained with both members and non-members using their output from the unlearned model $\{\mathcal{F}'(\mathbf{x}) \,|\, \mathbf{x} \in \mathcal{D}^M \cup \mathcal{D}^N\}$ as features and labels as $\{\mathbf{y}_i \,|\, \mathbf{y}_i = 1 \; \forall x_i \in \mathcal{D}^M, \mathbf{y}_i = 0 \; \forall x_i \in \mathcal{D}^N\}$. We then test the attack model on the unlearning samples $\mathcal{D}^u_{tr}$ and selected test member samples from remaining samples $\mathcal{D}^r_{tr}$. We report the **Member prediction rate** defined as number of positive (member) predictions by the MIA divided by total number of tests. It can be considered as false positive rate (FPR) for unlearning samples (considering them as non-members) and true positive rate (TPR) for members. An effective unlearning algorithm should have a low member prediction rate on unlearning samples and high member prediction rate on member samples. Our metric is consistent with existing literature Jia et al. (2023) utilizing true negative rate (TNR) for unlearning samples and test non-member samples (considering both as non-members), which essentially measures the opposite to ours, i.e., considering non-members rather than members. We focus on predicting the members because MIA is designed to infer members.

## 5.2 RESULTS ON SINGLE CLASS UNLEARNING

Table 1: Performance evaluation for single class unlearning on CIFAR-10.

| Model | Evaluation | Retrain (reference) | **Contrastive** | Boundary Shrink | Boundary Expansion | SCRUB | UNSIR |
|---|---|---|---|---|---|---|---|
| RN18 | Remain test↑ | 86.96 | **85.79** | 83.62 | 82.34 | 83.91 | 57.36 |
| | Unlearn train↓ | 0.00 | 0.00 | 4.54 | 0.00 | 35.42 | 0.00 |
| | Unlearn test↓ | 0.00 | 0.00 | 4.62 | 6.51 | 9.30 | 0.00 |
| RN34 | Remain test↑ | 88.01 | **86.59** | 84.70 | 83.19 | 82.22 | 47.02 |
| | Unlearn train↓ | 0.00 | 0.00 | 2.46 | 0.00 | 3.18 | 0.00 |
| | Unlearn test↓ | 0.00 | 0.00 | 4.60 | 6.81 | 0.80 | 0.00 |
| RN50 | Remain test↑ | 87.78 | **87.98** | 85.52 | 83.39 | 84.44 | 37.41 |
| | Unlearn train↓ | 0.00 | 0.00 | 2.74 | 0.00 | 7.16 | 0.00 |
| | Unlearn test↓ | 0.00 | 0.00 | 5.90 | 8.22 | 1.51 | 0.00 |
| RN101 | Remain test↑ | 87.94 | **88.69** | 83.91 | 82.48 | 85.03 | 42.40 |
| | Unlearn train↓ | 0.00 | 0.00 | 4.91 | 0.00 | 13.46 | 0.00 |
| | Unlearn test↓ | 0.00 | 0.00 | 7.25 | 8.50 | 4.55 | 0.00 |
| ViT | Remain test↑ | 75.56 | **70.63** | 69.36 | 40.36 | 68.26 | 24.43 |
| | Unlearn train↓ | 0.00 | 0.00 | 0.00 | 0.00 | 0.00 | 0.00 |
| | Unlearn test↓ | 0.00 | 0.00 | 0.00 | 0.00 | 0.00 | 0.00 |

**Unlearning Efficacy and Model Performance.** Table 1 depicts accuracy of different unlearned models on remain test (test set of remaining classes), unlearn train (train set of unlearning class), and unlearn test (test set of unlearning class) on CIFAR-10 for class 5. We experimented with all classes and they show similar performances. Readers may refer to the appendix. The retrain model shows the expected results with stable accuracy on remain test set (similar to the accuracy of original models shown in the Appendix) and zero for both unlearn train and unlearn test sets since the class has been removed from training. Among all methods, contrastive unlearning is the only one that achieves zero accuracy on the unlearning class indicating complete unlearning while preserving the accuracy on the remained classes. In fact, the unlearn test accuracy of contrastive unlearning reached very fast to zero, and by the time the termination condition was first checked, the unlearn

Table 2: Processing time of class unlearning algorithms on CIFAR-10 dataset (seconds).

| Model | Retrain | **Contrastive** | Boundary Shrink | Boundary Expansion | SCRUB | UNSIR |
|-------|---------|-----------------|-----------------|--------------------|-------|-------|
| RN18  | 1566.36 | **48.90**  | 105.22  | 112.87  | 150.40  | 59.98   |
| RN34  | 2072.76 | **75.45**  | 181.12  | 139.90  | 240.39  | 90.58   |
| RN50  | 3820.62 | **105.41** | 315.69  | 240.44  | 435.49  | 169.89  |
| RN101 | 7493.79 | **139.94** | 540.21  | 425.77  | 747.65  | 270.38  |
| ViT   | 22888.08| **256.12** | 2130.60 | 1950.72 | 1891.14 | 1525.92 |

Table 3: Performance evaluation on sample unlearning on CIFAR-10.

| Model | Evaluation | Retrain (reference) | **Contrastive** | Finetune | Gradient Ascent | Fisher | LCODEC |
|-------|-----------|---------------------|-----------------|----------|-----------------|--------|--------|
| RN18  | Test acc↑    | 84.68±0.23 | **81.86±0.33** | 81.68±0.29 | 67.64±3.41 | 76.54±2.34 | 76.20±1.37 |
|       | Unlearn acc  | 85.30±0.6  | 81.69±0.24     | 83.65±2.5  | 88.65±3.86 | 92.83±2.71 | 99.65±0.24 |
|       | Unlearn score↓ | 0.62     | 0.17           | 1.97       | 21.01      | 16.29      | 23.45      |
| RN34  | Test acc↑    | 85.48±0.14 | **83.53±0.54** | 82.38±0.80 | 67.54±3.41 | 76.54±2.34 | 81.22±0.85 |
|       | Unlearn acc  | 85.12±0.21 | 81.50±1.4      | 82.7±0.89  | 88.65±3.86 | 92.85±2.73 | 99.53±0.23 |
|       | Unlearn score↓ | 0.08     | 2.03           | 0.32       | 12.11      | 16.31      | 18.31      |
| RN50  | Test acc↑    | 86.44±0.57 | **84.80±0.34** | 82.60±0.51 | 67.70±5.22 | 72.03±8.00 | 78.14±1.04 |
|       | Unlearn acc  | 86.86±0.52 | 83.20±0.00     | 82.46±1.59 | 91.80±1.12 | 85.15±12.1 | 99.31±0.45 |
|       | Unlearn score↓ | 0.42     | 1.6            | 0.14       | 24.10      | 13.12      | 21.17      |
| RN101 | Test acc↑    | 85.98±0.13 | **86.75±0.87** | 83.76±1.16 | 76.76±6.71 | 82.81±0.83 | 78.62±1.11 |
|       | Unlearn acc  | 86.11±0.27 | 85.34±0.87     | 82.23±1.58 | 94.18±3.34 | 98.30±0.93 | 99.08±0.78 |
|       | Unlearn score↓ | 0.31     | 1.41           | 0.53       | 17.42      | 15.49      | 20.46      |
| ViT   | Test acc↑    | 73.28±0.52 | 62.02±0.49     | 73.08±2.35 | 69.25±3.17 | 20.66±3.10 | **84.54±0.78** |
|       | Unlearn acc  | 73.40±0.82 | 59.67±0.90     | 96.43±3.23 | 95.93±2.59 | 24.98±3.30 | 89.23±0.97 |
|       | Unlearn score↓ | 0.12     | 2.35           | 23.35      | 26.68      | 4.32       | 4.69       |

test accuracy had already dropped to zero. Readers may refer to Appendix D.3 for more details. UNSIR is the only baseline achieving 0 accuracy in the unlearning class, however, it suffers from a significant performance loss. All other methods fail to completely remove the influence while also showing a performance loss in the remaining classes.

**Efficiency.** Table 2 shows the elapsed time for each unlearning algorithm. Contrastive unlearning is the fastest among all baselines and across all models because it only requires running a single iteration over unlearning samples. The speed of UNSIR is second fastest as it also runs for a single iteration; however, extra time is consumed computing adequate noise to perturb parameters.

## 5.3 RESULTS ON SAMPLE UNLEARNING

**Model Performance and Unlearning Efficacy.** Table 3 shows accuracy on unlearning samples and test samples on the CIFAR-10 dataset. Successful sample unlearning should achieve high test accuracy (model utility) and an unlearn accuracy no higher than test accuracy with a low unlearn score (unlearning efficacy). The retrain models, which are the reference for unlearning, prove this point as they exhibit high test accuracy and unlearn score close to 0. Contrastive unlearning is best performing among all methods, achieving the closest performance to the retrain model. While finetune shows a smaller unlearn score than contrastive unlearning for some models, the difference is negligible and it has lower test accuracy on these models. In addition, it completely fails to unlearn on ViT (with an unlearn accuracy much higher than test accuracy).

**Unlearning Efficacy via MIA.** Table 4 shows the member prediction rate of the MIA on unlearning samples and test member samples against each unlearned model. An ideal attack model against the retrain model should have zero member prediction rate for unlearning samples and 100% for member samples (since the unlearning samples are non-members). However, the attack model in our experiment shows around 60% for unlearning samples on the retrain model, which is due to

Table 4: Member prediction rate on unlearning samples (lower the better) and member-test samples (memorized train samples) of MIA on CIFAR-10 dataset.

| Model | Evaluation | Retrain (reference) | **Contrastive** | Finetune | Gradient Ascent | Fisher | LCODEC |
|---|---|---|---|---|---|---|---|
| RN18 | unlearning↓ | 63.28±0.48 | **60.88±0.78** | 63.87±0.98 | 79.85±1.13 | 85.91±1.26 | 92.18±1.41 |
| | member-test | 96.08±0.52 | 91.05±0.59 | 85.81±1.01 | 84.62±1.12 | 89.23±1.31 | 92.98±0.89 |
| RN34 | unlearning↓ | 63.81±0.55 | **53.51±0.58** | 66.65±0.87 | 83.08±0.99 | 82.59±1.10 | 95.49±1.13 |
| | member-test | 94.82±0.32 | 86.44±0.46 | 86.99±0.84 | 84.01±1.18 | 83.74±0.98 | 97.21±1.21 |
| RN50 | unlearning↓ | 63.04±0.29 | **60.87±0.64** | 68.47±0.89 | 85.87±1.08 | 74.46±1.42 | 93.98±1.35 |
| | member-test | 97.43±0.47 | 91.13±0.54 | 84.03±0.93 | 89.29±1.29 | 77.15±1.68 | 93.59±1.56 |
| RN101 | unlearning↓ | 62.49±0.51 | 60.79±0.78 | **54.89±0.99** | 91.98±1.14 | 84.20±1.86 | 94.93±1.53 |
| | member-test | 95.74±0.62 | 86.45±0.92 | 62.39±1.05 | 90.47±0.89 | 84.90±1.77 | 95.10±1.68 |
| ViT | unlearning↓ | 53.57±0.38 | **55.49±0.74** | 84.97±1.04 | 56.58±1.23 | 56.18±1.59 | 83.99±1.48 |
| | member-test | 89.29±0.76 | 72.87±0.69 | 85.92±1.18 | 57.49±1.44 | 59.86±0.88 | 87.12±1.43 |

Table 5: Processing time of algorithms conducting sample unlearning on CIFAR-10 (minutes)

| Model | Retrain | **Contrastive** | Finetune | Gradient Ascent | Fisher | LCODEC |
|---|---|---|---|---|---|---|
| RN18 | 43.05±2.18 | **2.68±0.64** | 16.93±2.24 | 4.89±0.82 | 72.31±1.52 | 34.87±1.87 |
| RN34 | 73.22±3.44 | **3.64±0.72** | 31.51±2.21 | 7.52±1.21 | 115.51±1.98 | 55.50±1.15 |
| RN50 | 134.42±4.72 | **8.46±0.98** | 42.93±3.52 | 14.16±1.46 | 219.49±1.95 | 152.28±1.64 |
| RN101 | 215.84±4.57 | **12.63±1.02** | 103.74±3.05 | 20.21±1.41 | 398.87±1.66 | 449.11±1.31 |
| ViT | 402.15±3.73 | **3.10±0.45** | 79.24±3.61 | 35.65±1.19 | 218.93±1.48 | 1719.60±3.41 |

the attack power of the attack model. The high rate on member samples suggests it has reasonable attack power in recognizing members. We expect stronger attack methods Carlini et al. (2022) can better differentiate members and non-members but the comparison of the methods should stay the same. An unlearning algorithm is more effective if it exhibits 1) lower member prediction rate on unlearning samples, and 2) bigger difference in member prediction rate on unlearning samples and member samples. For gradient ascent, Fisher, and LCODEC, the member prediction rate for member samples and unlearning samples are similar, showing ineffective unlearning. For finetune and contrastive unlearning, the member prediction rate for unlearning samples is lower than member samples. However, the difference is significantly bigger in contrastive unlearning, suggesting stronger discrimination between unlearning samples and member samples and more effective unlearning.

**Efficiency.** Table 5 shows the runtime of different algorithms. It shows contrastive unlearning is the fastest to reach the termination condition. On average, it needed less than 15 unlearning rounds, which is computation equivalent to at most $15 \times \omega$ iterations on unlearning dataset. While gradient ascent also iterates only on unlearning dataset, it requires more than 40 iterations to achieve unlearning effects, and requires a smaller batch size for the better results. Finetune, Fisher, and LCODEC need longer runtime since they iterate over the entire set of remaining samples. Fisher and LCODEC suffer excessive computation with larger models because their mathematical computation is proportional to model parameters and hardly parallelizable.

## 6 Conclusion

In this paper, we proposed a novel contrastive approach for machine unlearning. It achieves unlearning by re-configuring geometric properties of embedding space and contrasting unlearning samples and remaining samples. Through extensive experiments, we demonstrated that it outperforms state-of-the-art unlearning algorithms in model performance, unlearning efficacy, and efficiency. In future work, we will examine the efficacy of contrastive unlearning in different model architectures and different unlearning scenarios such as graph unlearning and correlated sequence unlearning.

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

## A    APPENDIX / SUPPLEMENTAL MATERIAL

In this appendix, Section B illustrates full algorithm of our contrastive unlearning. Section C provides details on the implementation of our contrastive unlearning, a link to the implementation (code), and a list of hyperparameters used for experiments. Section D provides the results of additional experiments on contrastive unlearning. We provide additional experiments on class unlearning, the efficiency and effectiveness of the unlearning SVHN and Mini-Imagenet dataset and an ablation- study on hyperparameters.

## B    ALGORITHM

---

**Algorithm 1** Contrastive Unlearning

---

**Input**: $\theta, \mathcal{H}(\cdot), \mathcal{E}(\cdot), D_{tr}^r, D_{tr}^u, \mathcal{D}_{\text{eval}}$
**Parameter**: $iter, \lambda_{CL}, \lambda_{UL}, \omega$
**Output**: $\theta'$

1: **while** termination condition is not satisfied **do**
2:     **for** each $X^u \in D_{tr}^u$ **do**
3:         **for** $1, \cdots, \omega$ **do**
4:             Sample $(X^r, Y^r)$ from $\mathcal{D}_{tr}^r$
5:             Determine $P_{\mathbf{z}}(x_i), N_{\mathbf{z}}(x_i) \, \forall x_i \in X^u$
6:             $\ell_{CE} \leftarrow \mathcal{L}_{CE}(\mathcal{H}(\mathcal{E}_\theta(X^r)), Y^r)$
7:             $\ell_{UL} \leftarrow \lambda_{UL}\mathcal{L}_{UL}(P_{\mathbf{z}}(x_i), N_{\mathbf{z}}(x_i)) \, \forall x_i \in X^u$
8:             $\theta \leftarrow \theta - \eta\nabla(\ell_{CE} + \ell_{UL})$
9:         **end for**
10:     **end for**
11:     $\theta' \leftarrow \theta$
12:     Evaluate, get termination condition $\theta'$ with $\mathcal{D}_{\text{eval}}$
13: **end while**
14: **return** $\theta'$

---

**Complete Algorithm.** Algorithm 1 shows step-wise overview of contrastive unlearning. It iterates for all unlearning batches $X^u$ in $D_{tr}^u$. For each $X^u$, it computes unlearning loss by sampling a random remaining batch $X^r$ for contrasting purposes. For each $X^u$, sampling and loss derivation are repeated $\omega$ times. Higher $\omega$ stabilizes the unlearning procedure by contrasting unlearning samples against multiple sets of remaining samples. From the experiment, we set $\omega$ to be at most 4 to reduce computational overhead and our algorithm showed stable unlearning performance.

## C    EXPERIMENTAL DETAILS

Our implementation is based on PyTorch Paszke et al. (2019). We used one Quadro RTX 8000 with memory size of 48,600 MB. Our code is available on an anonymous git repository.

For ResNet and ViT models on CIFAR-10 and SVHN dataset, we used these hyperparamters. We used stochastic gradient descent for training ResNet models and Adam optimizer for training ViT

Table 6: Hyperparameters for the CIFAR-10 and SVHN datasets.

| | CIFAR-10 | | SVHN | |
|---|---|---|---|---|
| Hyperparameter | Sample Unlearn | Class Unlearn | Sample Unlearn | Class Unlearn |
| Feature dimension | 128 | 128 | 128 | 128 |
| Batch size | 128 | 64 | 128 | 64 |
| $\lambda_{CE}$ | 1 | 1 | 2 | 1 |
| $\lambda_{UL}$ | 3 | 3 | 3 | 3 |
| $\omega$ | 4 | 4 | 4 | 4 |
| $\tau$ | 0.7 | 0.7 | 0.7 | 0.7 |
| Learning rate | $1e^{-3}$ | $1e^{-3}$ | $1e^{-3}$ | $1e^{-3}$ |
| Weight decay | $5e^{-4}$ | $5e^{-4}$ | $5e^{-4}$ | $5e^{-4}$ |
| Momentum | 0.9 | 0.9 | 0.9 | 0.9 |

Table 7: Hyperparameters for Mini-Imagenet dataset. We used same hyperparameter settings for both class and sample unlearning

| Hyperparameter | ResNet18 | ResNet34 | ResNet50 | ResNet101 |
|---|---|---|---|---|
| Feature dimension | 256 | 256 | 512 | 512 |
| Batch size | 256 | 128 | 128 | 128 |
| $\lambda_{CE}$ | 1 | 1 | 2 | 1 |
| $\lambda_{UL}$ | 3 | 3 | 3 | 3 |
| $\omega$ | 4 | 4 | 4 | 4 |
| $\tau$ | 0.7 | 0.7 | 0.7 | 0.7 |
| Learning rate | $1e^{-3}$ | $1e^{-3}$ | $1e^{-4}$ | $1e^{-4}$ |
| Weight decay | $5e^{-4}$ | $5e^{-4}$ | $5e^{-4}$ | $5e^{-4}$ |
| Momentum | 0.9 | 0.9 | 0.9 | 0.9 |

# D    ADDITIONAL EXPERIMENTS

## D.1    PERFORMANCE OF ORIGINAL MODELS

We use three standard benchmark datasets, CIFAR-10 Krizhevsky et al. (2009) and SVHN Netzer et al. (2011) and Mini-imagenet Cao (2022). The original mini-imagenet is designed for few-shot learning Vinyals et al. (2016) so its distribution makes training a model from scratch difficult. Instead, we used a modified version whose distribution is adjusted for image classification task Cao (2022). For models, we used ResNet-18, 34, 50, and 101 models and ViT in our experiments. We train each model with each dataset. For CIFAR-10 and SVHN, we trained the models without any data augmentation except normalization. For Mini-Imagenet, we used image augmentation techniques such as RandomRotation and RandomCrop. The performance of each original model is shown in Table 8. We then apply unlearning algorithms to the trained models. We did not train ViT against Mini-Imagenet dataset because training ViT with small dataset is difficult and often leads poor performance Liu et al. (2021).

Table 8: Performance of original models.

| Dataset | | RN18 | RN34 | RN50 | RN101 | ViT |
|---|---|---|---|---|---|---|
| CIFAR-10 | Train | 100.0 | 100.0 | 100.0 | 100.0 | 100.0 |
| | Test | 85.81 | 86.62 | 87.5.0 | 86.69 | 72.72 |
| SVHN | Train | 99.98 | 99.88 | 99.99 | 99.84 | 100.0 |
| | Test | 95.32 | 95.86 | 95.94 | 96.14 | 87.81 |
| Mini-Imagenet | Train | 96.07 | 96.07 | 97.03 | 97.03 | - |
| | Test | 68.19 | 68.18 | 71.81 | 72.57 | - |

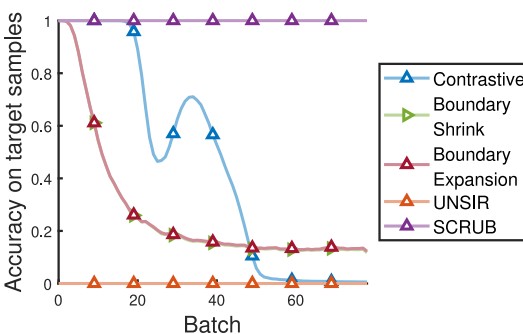

Figure 2: Accuracy on unlearning class vs. number of batches on $\mathcal{D}_{tr}^u$.

## D.2 Unlearning Each Class

For single class unlearning, we reported results for unlearning class 5 from CIFAR-10 and SVHN dataset. We also experimented with unlearning different classes which verified the effectiveness of contrastive unlearning. Table 9 and 10 show accuracy of unlearned models on $\mathcal{D}_{ts}^r$ (test set of remaining classes), $\mathcal{D}_{tr}^u$ (train set of unlearning class), and $\mathcal{D}_{ts}^u$ (test set of unlearning class) on CIFAR-10 and SVHN respectively. The table clearly shows that contrastive unlearning is capable of unlearning each class as accuracy of test set and train set of unlearning class are all zero, indicating that each model is capable of removing influence completely. At the same time, the accuracy of test set of remaining classes is preserved and similar to the original model.

Table 9: performance evaluation for unlearning each class of CIFAR-10 dataset

| Unlearning Class | $\mathcal{D}_{ts}^r$ | $\mathcal{D}_{tr}^u$ | $\mathcal{D}_{ts}^u$ |
|---|---|---|---|
| 0 | 84.97 | 0.00 | 0.00 |
| 1 | 84.62 | 0.00 | 0.00 |
| 2 | 85.18 | 0.00 | 0.00 |
| 3 | 86.38 | 0.00 | 0.00 |
| 4 | 84.73 | 0.00 | 0.00 |
| 5 | 85.79 | 0.00 | 0.00 |
| 6 | 83.07 | 0.00 | 0.00 |
| 7 | 83.71 | 0.00 | 0.00 |
| 8 | 83.92 | 0.00 | 0.00 |
| 9 | 85.03 | 0.00 | 0.00 |

Table 10: performance evaluation for unlearning each class of SVHN dataset

| Unlearning Class | $\mathcal{D}_{ts}^r$ | $\mathcal{D}_{tr}^u$ | $\mathcal{D}_{ts}^u$ |
|---|---|---|---|
| 0 | 93.98 | 0.00 | 0.00 |
| 1 | 94.31 | 0.00 | 0.00 |
| 2 | 94.20 | 0.00 | 0.00 |
| 3 | 94.57 | 0.00 | 0.00 |
| 4 | 94.11 | 0.00 | 0.00 |
| 5 | 93.81 | 0.00 | 0.00 |
| 6 | 94.09 | 0.00 | 0.00 |
| 7 | 94.12 | 0.00 | 0.00 |
| 8 | 93.93 | 0.00 | 0.00 |
| 9 | 93.91 | 0.00 | 0.00 |

## D.3 Efficiency of Class unlearning

Figure 2 shows the progress of the unlearning algorithms in terms of the accuracy on unlearning class $\mathcal{D}_{tr}^u$ vs. the number of batches in a single epoch. Both contrastive unlearning and other baselines are designed to run unlearning procedures multiple times for each batch. However, we fixed the hyperparameters of each algorithm so that each batch of $\mathcal{D}_{tr}^u$ is processed only once. Reaching faster to zero accuracy indicates that the algorithm is more efficient, as it needs a smaller number of batches to achieve unlearning. The figure shows that contrastive unlearning reaches zero approximately at the 60th batch while boundary shrink and boundary expansion still show approximately 10% accuracy after the first epoch. UNSIR shows zero accuracy from the beginning. However, it computes the proper level of noise by iterating through $\mathcal{D}_{tr}^u$ before running actual optimization. SCRUB, which is based on knowledge distillation, requires several passes through the $\mathcal{D}_{tr}^u$ and hence does not show any progress after one epoch. In summary, contrastive unlearning is most efficient as it achieves unlearning by only requiring 60 batches to achieve unlearning.

## D.4 UNLEARNING LARGE NUMBER OF SAMPLES

For random sample unlearning, we compared the unlearn efficacy and performance of the model from unlearning 500 randomly selected samples. We also experimented unlearning randomly selected 250, 500, 1000 and 2000 samples to show the robustness of contrastive unlearning against the baselines. Table 11 shows the result of unlearning various number of samples. It shows that both contrastive unlearning and Fisher unlearning suffers utility loss as number of unlearning sample increases. However, contrastive unlearning suffers smaller performance loss. With unlearning 2000 samples, it suffers about 8% of test accuracy. On the other hand, fisher unlearning suffers significant performance loss. Its test accuracy becomes random guess on unlearning 2000 samples. This shows that the contrastive unlearning is capable of unlearning larger number of samples.

Table 11: Random sample unlearning with various number of unlearning size

|  | Retrain | Contrastive Unlearning | | Fisher Unlearning | |
| --- | --- | --- | --- | --- | --- |
| Unlearn Size | Test acc ↑ | Test acc↑ | Unlearn Acc | Test acc↑ | Unlearn Acc |
| 200 | 86.38 | 82.30 | 76.40 | 77.48 | 98.00 |
| 500 | 86.32 | 82.15 | 81.60 | 77.40 | 96.00 |
| 1000 | 85.71 | 82.15 | 81.66 | 40.78 | 50.00 |
| 2000 | 84.95 | 76,39 | 76.35 | 10.94 | 15.20 |

While contrastive unlearning is capable of removing influence of larger number of unlearning samples, it impairs the performance of the model. Therefore, the number of unlearning samples should be limited by the maximum performance loss the system is able to tolerate.

## D.5 EFFECT OF HYPERPARAMETER $\tau$

For every experiment, we set $\tau = 0.7$ to follow default setting of supervised contrastive learning Khosla et al. (2020). Hence in this section we report the effect of various $\tau$. Table 12 shows the unlearn efficacy and model performance on various $\tau$. It shows that $\tau$ does not have a significant impact on the unlearn and test accuracy. One thing we noticed is that the smaller $\tau$ slightly increases the difference between test and unlearn accuracy.

Table 12: Test Accuracy, Unlearn Accuracy, and Time for various $\tau$ values

| $\tau$ | Test acc. | Unlearn acc. | Time (seconds) |
| --- | --- | --- | --- |
| 0.007 | 82.20 | 79.40 | 134.57 |
| 0.07 | 82.12 | 80.20 | 121.66 |
| 0.7 | 82.15 | 81.60 | 109.32 |
| 7 | 82.15 | 81.60 | 111.61 |
| 70 | 82.15 | 81.60 | 115.86 |

## D.6 UNLEARNING FEW-SHOT CLASSIFIER

Unlike other baseline algorithms, contrastive unlearning modifies embeddings of unlearning samples to achieve unlearning. It implies that contrastive unlearning is capable of unlearning models beyond the standard classification models such as vision language models learned through contrastive learning. To verify this claim, we conduct an experiment on unlearning CLIP model Radford et al. (2021). The CLIP is pretrained with large number of image and text pairs. Since the original data is publicly unavailable, we first finetune the pretrained model with CIFAR-100 dataset for 10 epochs. The finetuned model achieved top-1 accuracy of 82.3%. Then we attempted to unlearn a class from the finetuned model. Similar to the class unlearning problem, we unlearned all samples of a target class until it reaches the accuracy of random guess. We do not compare the results with other baselines except for Finetune and Gradient Ascent since these baselines are designed to only handle standard classification models that provide prediction logits. Hence they are unable to unlearn

CLIP. For Finetune, we further finetune the CLIP only with samples of remaining class to accelerate catastrophic forgetting of the unlearning samples. For Gradient Ascent, we conduct gradient ascent for the unlearning samples using the contrastive loss, and conduct gradeint descent for retaining samples with the same loss.

Table 13: Performance evaluation on class unlearning on CLIP

| Method | Unlearn acc. | Test acc |
|---|---|---|
| Contrastive | 65.00 | 0.0 |
| Gradient ascent | 12.42 | 0.0 |
| Finetune | 79.87 | 87.00 |

Table 13 shows the result of unlearning class 1 of CIFAR-100 dataset from CLIP. It shows that contrastive unlearning was capable of achieving good unlearning utility as the model exhibits classification accuracy below random guess for the target samples. While Gradient Ascent was able to achieve similar unlearning effect, the performance loss is significant compared with contrastive unlearning. While Finetune was able to preserve the model utility, the result shows that unlearn efficacy is not good since its unlearn accuracy is significantly higher than random guess. The results show that contrastive unlearning is able to achieve good unlearn efficacy with small performance loss.

### D.7 SCALABILITY: USING ADVANCED CONTRASTING TECHNIQUES FOR CONTRASTIVE UNLEARNING

From section 4, we illustrate the concept of contrastive unlearning using supervised contrastive learning Khosla et al. (2020). Within a batch, contrastive unlearning pulls unlearning samples' embeddings towards the remaining samples with different class and pushes the unlearning samples' embeddings away from the remaining samples with the same class. Since our default implementation is based on the supervised contrastive learning (SupCon), it inherits its weaknesses. A critical problem of SupCon is that it requires extensive batch size. Since each sample in a batch is only contrasted with samples within the batch, having smaller batch size increases bias to directions where each samples are optimized. To reduce bias and facilitate stable representation learning, SupCon requires larger batch size. In our contrastive unlearning, we also experienced that unlearning becomes very instable for smaller batch size and reported relevant explanation in Appendix D.10.

These problems can be effectively mitigated via adopting more stable contrastive learning algorithms using the same contrastive unlearning principle. To empirically show this, we implemented contrastive unlearning using Momentum Contrast (MoCo) He et al. (2020). From MoCo, the contrastive loss for embeddings of a sample $z$ is defined as follows:

$$\mathcal{L} = -\log \frac{\exp\left(z \cdot z_+/\tau\right)}{\sum_i^K \exp\left(z \cdot z_i/\tau\right)} \tag{10}$$

The loss is pulling $z$ towards a positive sample $z_+$, and pushing $z$ away from $K$ negative samples. In MoCo, these $k$ negative samples are stored in a queue to mitigate introducing bias from the batch size. Intuitively, it can be seen as a softmax-based classifier with $K+1$ classes. By slightly modifying the loss, we can achieve contrastive unlearning.

$$\mathcal{L}_{UL} = -\log \frac{\sum_i^J \exp\left(z \cdot z_i^+/\tau\right)}{\sum_i^K \exp\left(z \cdot z_i^-/\tau\right)} \tag{11}$$

where $z_i^+$ are embeddings of remaining samples with different class, and $z_i^-$ are the embeddings of samples with same class. Similar to MoCo, $z_i^+$ and $z_i^-$ are stored within a queue. We conduct sample unlearning from ResNet-18 model using MoCo based implementation.

Table 14: Performance evaluation of Sample Unlearning using Momentum Contrastive algorithm

| Unlearn acc. | Test acc |
|---|---|
| 76.45 | 71.80 |

Table 14 shows the result of MoCo based contrastive unlearning. It shows that our contrastive unlearning framework is not confined to a particular contrastive learning technique and it can be effectively implemented via more advanced contrastive learning techniques. We deem that effectiveness of different contrastive learning technique depends on the structure and the size of the dataset. We plan to provide further insight in future research.

## D.8 SVHN DATASET

### D.8.1 SINGLE CLASS UNLEARNING ON SVHN DATASET

Table 15: Performance evaluation for single class unlearning on SVHN.

| Model | Evaluation | Retrain (reference) | **Contrastive** | Boundary Shrink | Boundary Expansion | SCRUB | UNSIR |
|---|---|---|---|---|---|---|---|
| RN18 | Remain test↑ | 95.43 | 93.91 | 94.84 | 93.71 | 93.88 | 90.3 |
| | Unlearn train↓ | 0.00 | 0.00 | 29.79 | 80.25 | 88.67 | 0.00 |
| | Unlearn test↓ | 0.00 | 0.00 | 37.46 | 2.61 | 77.39 | 0.00 |
| RN34 | Remain test↑ | 95.46 | 94.33 | 95.12 | 94.50 | 94.57 | 85.82 |
| | Unlearn train↓ | 0.00 | 0.00 | 34.69 | 63.92 | 0.96 | 0.00 |
| | Unlearn test↓ | 0.00 | 0.00 | 41.99 | 4.27 | 0.42 | 0.00 |
| RN50 | Remain test↑ | 95.83 | 94.87 | 95.47 | 95.01 | 93.75 | 70.56 |
| | Unlearn train↓ | 0.00 | 0.00 | 40.01 | 3.92 | 2.68 | 0.00 |
| | Unlearn test↓ | 0.00 | 0.00 | 42.37 | 8.74 | 9.64 | 0.00 |
| RN101 | Remain test↑ | 96.16 | 94.90 | 95.65 | 95.07 | 94.65 | 83.90 |
| | Unlearn train↓ | 0.00 | 0.00 | 42.77 | 51.53 | 0.00 | 0.00 |
| | Unlearn test↓ | 0.00 | 0.00 | 45.39 | 3.94 | 0.00 | 0.00 |
| ViT | Remain test↑ | 87.78 | 77.45 | 65.33 | 14.63 | 21.99 | 87.66 |
| | Unlearn train↓ | 0.00 | 0.00 | 0.00 | 0.00 | 0.00 | 6.16 |
| | Unlearn test↓ | 0.00 | 0.00 | 2.14 | 0.00 | 0.00 | 0.00 |

Table 15 illustrates accuracy of unlearned models on SVHN dataset. It shows a similar trend as the CIFAR-10 dataset. UNSIR provides better performance on the SVHN dataset because features of SVHN are easier to learn thus the model suffers less utility loss than CIFAR-10. However, it still suffers a significantly higher utility loss than contrastive unlearning. All other baselines show a high accuracy on the unlearning class in many cases, indicating they failed to remove the influence of the unlearning class. Contrastive unlearning consistently removed all influence of unlearning class with a negligibly small loss of performance.

### D.8.2 SAMPLE UNLEARNING ON SVHN DATASET

Table 16 presents test and unlearning accuracy on the SVHN dataset. LCODEC and Fisher show similar test accuracy with the retrain model on some models. However, their unlearning accuracy is very high, at almost 100%, indicating a significant residual of the influence. Both Finetune and gradient ascent show significant performance loss in test accuracy. Contrastive unlearning is more consistent in achieving similar unlearning accuracy as the retrain model with a relatively small performance loss in test accuracy.

Table 16: Performance evaluation on sample unlearning on SVHN.

| Model | Evaluation | Retrain | **Contrastive** | Finetune | Gradient Ascent | Fisher | LCODEC |
|---|---|---|---|---|---|---|---|
| RN18 | Test acc↑ | 94.89±0.21 | 91.67±0.29 | 91.66±0.35 | 67.80±16.8 | 88.76±1.64 | 93.49±1.09 |
| | Unlearn acc | 94.20±0.13 | 90.35±0.57 | 90.85±0.1 | 96.9±2.14 | 97.55±2.04 | 99.63±0.49 |
| | Unlearn score↓ | 0.69 | 0.82 | 0.81 | 29.1 | 8.79 | 6.14 |
| RN34 | Test acc↑ | 95.39±0.32 | 93.01±0.15 | 92.52±0.58 | 84.03±7.91 | 91.25±0.59 | 94.95±1.19 |
| | Unlearn acc | 94.12±0.14 | 91.50±0.60 | 90.90±0.90 | 97.65±1.45 | 97.00±0.84 | 99.48±0.49 |
| | Unlearn score↓ | 1.27 | 1.51 | 1.60 | 12.72 | 5.75 | 4.53 |
| RN50 | Test acc↑ | 95.86±0.25 | 93.50±0.25 | 93.01±0.81 | 71.47±20.8 | 91.46±0.05 | 94.46±0.71 |
| | Unlearn acc | 95.12±0.47 | 92.75±0.41 | 92.00±1.12 | 96.73±3.66 | 97.80±0.00 | 99.48±0.53 |
| | Unlearn score↓ | 0.74 | 0.75 | 1.01 | 25.26 | 6.34 | 5.02 |
| RN101 | Test acc↑ | 95.88±0.22 | 92.89±0.46 | 91.98±0.39 | 78.35±8.23 | 94.25±0.81 | 82.42±1.03 |
| | Unlearn acc | 93.45±0.78 | 91.29±0.87 | 91.00±0.1 | 97.30±5.27 | 99.80±0.00 | 92.87±0.66 |
| | Unlearn score↓ | 2.21 | 1.60 | 0.98 | 18.95 | 5.55 | 10.45 |
| ViT | Test acc↑ | 86.45±0.18 | 73.28±0.39 | 86.23±0.79 | 21.42±8.24 | 6.29±0.52 | 86.28±0.97 |
| | Unlearn acc | 85.35±0.62 | 72.20±0.72 | 98.92±0.58 | 68.12±6.28 | 8.87±0.13 | 99.82±0.42 |
| | Unlearn score↓ | 1.10 | 1.08 | 12.69 | 46.7 | 2.58 | 13.54 |

### D.8.3 EFFICIENCY OF CLASS UNLEARNING ON SVHN DATASET

We reported efficiency of class unlearning on CIFAR-10 dataset to show contrastive unlearning is the most efficient framework. Similarly, here we provide efficiency analysis of class unlearning on SVHN dataset. Table 17 shows the time required to unlearn each class using each framework. For a smaller model, SCRUB and UNSIR require less time; however, the effectiveness and performance of SCRUB and UNSIR are inferior to those of contrastive unlearning. With more complex models, baseline unlearning frameworks show sluggish computation. For ResNet101, the fastest baseline is UNSIR, which requires 990 seconds to run, while contrastive unlearning only requires 599 seconds.

Table 17: Processing time of class unlearning algorithms on SVHN dataset (in seconds).

| Model | Retrain | **Contrastive** | Boundary Shrink | Boundary Expansion | SCRUB | UNSIR |
|---|---|---|---|---|---|---|
| RN18 | 59007.60 | 519.44 | 1665.27 | 1620.27 | 480.39 | 407.28 |
| RN34 | 55404.20 | 568.37 | 1710.33 | 1646.22 | 604.56 | 810.42 |
| RN50 | 57276.10 | 597.95 | 1860.27 | 1665.30 | 900.42 | 901.02 |
| RN101 | 56822.40 | 599.42 | 2090.16 | 1695.30 | 1372.14 | 990.48 |
| ViT | 12201.84 | 1348.92 | 1650.60 | 1244.4 | 1374.36 | 701.1 |

### D.8.4 EFFICIENCY OF SAMPLE UNLEARNING ON SVHN DATASET

Table 18 shows the time required to unlearn randomly selected samples using each framework. Contrastive unlearning requires the lowest computation time. Finetune is faster than contrastive unlearning on ResNet34, and it is because of randomness within the algorithm. Fisher and LCODEC require extensive computation. LCODEC, specifically, is even slower than retraining.

Table 18: Processing time of sample unlearning algorithms on SVHN dataset (in minutes).

| Model | Retrain | **Contrastive** | Finetune | Gradient Ascent | Fisher | LCODEC |
|-------|---------|-----------------|----------|-----------------|--------|--------|
| RN18 | 515.83±0.87 | **43.48±0.24** | 199.28±1.98 | 51.69±1.25 | 121.16±0.03 | 418.01±0.77 |
| RN34 | 526.72±0.68 | **43.52±0.13** | 39.57±1.73 | 60.84±0.97 | 183.06±0.11 | 522.34±0.91 |
| RN50 | 538.14±0.59 | **41.09±0.28** | 368.03±1.49 | 82.68±0.99 | 301.57±0.14 | 938.39±0.86 |
| RN101 | 549.45±0.59 | **38.57±0.33** | 327.46±1.61 | 68.19±1.13 | 542.91±0.16 | 1918.87±0.91 |
| ViT | 192.54±0.34 | **2.05±0.41** | 35.03±0.99 | 4.08±1.18 | 203±0.14 | 1371.53±0.65 |

### D.8.5 EFFECTIVENESS (MIA) OF SAMPLE UNLEARNING ON SVHN DATASET

Table 19 shows the member prediction rate of the MIA on unlearning samples and test member samples. Contrast unlearning shows the lowest member prediction rate on unlearning samples and the biggest difference between the member prediction rate on unlearning samples and test member samples. While some baselines show a lower member prediction rate on unlearning samples, they present a very small difference between two member prediction rates. Some baselines show a low member prediction rate on test member samples. This does not directly indicate the corresponding unlearning framework is effective in unlearning. Instead, this is due to the technical limitations of the membership inference attack, and we aim to investigate more powerful MIA frameworks in future work.

| Model | Evaluation | Retrain | **Contrastive** | Finetune | Gradient Ascent | Fisher | LCODEC |
|-------|-----------|---------|-----------------|----------|-----------------|--------|--------|
| RN18 | unlearning↓ | 76.29±0.24 | 56.01±0.48 | 64.12±0.98 | 69.05±1.13 | 52.28± | 53.86±0.67 |
| | member-test | 83.10±0.39 | 74.14±0.37 | 64.78±0.82 | 75.01±1.22 | 59.86± | 59.43±0.86 |
| RN34 | unlearning↓ | 57.82±0.33 | 60.85±0.72 | 63.39±1.01 | 74.23±0.87 | 64.25± | 83.22±0.75 |
| | member-test | 63.27±0.41 | 76.83±0.68 | 63.98±0.96 | 77.83±1.05 | 66.34± | 81.71±0.88 |
| RN50 | unlearning↓ | 55.98±0.48 | 51.97±0.66 | 59.98±1.07 | 60.67±0.87 | 59.24± | 64.21±0.94 |
| | member-test | 64.97±0.58 | 61.49±0.59 | 63.94±0.93 | 64.18±1.25 | 60.62± | 68.49±0.98 |
| RN101 | unlearning↓ | 52.04±0.37 | 58.24±0.45 | 54.22±1.11 | 59.51±0.97 | 58.31± | 65.62±1.12 |
| | member-test | 57.99±0.51 | 73.66±0.56 | 60.17±1.02 | 58.89±1.33 | 55.61± | 64.72±1.33 |

Table 19: Member prediction rate on unlearning samples and member-test samples of MIA on SVHN dataset.

## D.9 MINI-IMAGENET DATASET

### D.9.1 SINGLE CLASS UNLEARNING ON MINI-IMAGENET DATASET

| Model | Evaluation | Retrain (reference) | **Contrastive** | Boundary Shrink | Boundary Expansion | SCRUB | UNSIR |
|---|---|---|---|---|---|---|---|
| RN18 | Remain test↑ | 65.62 | 60.69 | 10.17 | 51.26 | 50.20 | 17.05 |
| | Unlearn train↓ | 0.00 | 0.00 | 0.00 | 0.00 | 0.00 | 0.00 |
| | Unlearn test↓ | 0.00 | 0.00 | 0.00 | 0.95 | 0.00 | 0.00 |
| RN34 | Remain test↑ | 67.64 | 57.61 | 14.88 | 26.89 | 26.57 | 12.32 |
| | Unlearn train↓ | 0.00 | 0.00 | 0.00 | 0.00 | 0.00 | 0.00 |
| | Unlearn test↓ | 0.00 | 0.00 | 0.00 | 0.00 | 0.00 | 0.00 |
| RN50 | Remain test↑ | 70.57 | 58.81 | - | - | 22.03 | 12.74 |
| | Unlearn train↓ | 0.00 | 0.00 | - | - | 0.00 | 0.00 |
| | Unlearn test↓ | 0.00 | 0.00 | - | - | 0.00 | 0.00 |
| RN50 | Remain test↑ | 71.34 | 58.53 | - | - | 12.63 | 8.75 |
| | Unlearn train↓ | 0.00 | 0.00 | - | - | 0.00 | 0.00 |
| | Unlearn test↓ | 0.00 | 0.00 | - | - | 0.00 | 0.00 |

Table 20: Performance evaluation for single class unlearning on Mini-Imagenet dataset.

Table 20 shows the accuracy of unlearned models on Mini-Imagenet dataset. Similar to experiments on CIFAR-10 and SVHN dataset, re-trained model shows high test accuracy on remaining test classes, and zero accuracy for both test-set and train-set of unlearning class. Contrastive unlearning is most effective as it shows the highest classification accuracy on test-set of the remaining class. Unlike CIFAR-10 and SVHN datasets, contrastive unlearning suffers significant utility loss. We presume that it is due to the large number of classes. As mini-imagenet dataset has 100 classes, representation space might have intricate decision boundaries. Conducting contrastive unlearning could impair embeddings of remaining samples. We did not report experiments of Boundary Shrink and Boundary Expansion for ResNet50 and ResNet101 because they required excessive computational resource and produced out-of-memory error.

### D.9.2 SAMPLE UNLEARNING ON MINI-IMAGENET DATASET

| Model | Evaluation | Retrain | **Contrastive** | Finetune | Gradient Ascent | Fisher |
|---|---|---|---|---|---|---|
| RN18 | Test acc↑ | 66.17 | 54.40 | 69.53 | 45.61 | 11.67 |
| | Unlearn acc | 65.40 | 51.20 | 96.20 | 86.60 | 10.00 |
| | Unlearn score↓ | 1.87 | 3.2 | 26.67 | 40.99 | 1.67 |
| RN34 | Test acc↑ | 68.93 | 38.37 | 69.83 | 42.61 | 10.61 |
| | Unlearn acc | 66.60 | 37.20 | 96.20 | 86.60 | 18.00 |
| | Unlearn score↓ | 2.33 | 1.17 | 26.37 | 43.99 | 7.39 |
| RN50 | Test acc↑ | 71.26 | 55.71 | 72.69 | 52.05 | 11.67 |
| | Unlearn acc | 68.20 | 55.80 | 97.00 | 83.60 | 18.00 |
| | Unlearn score↓ | 3.06 | 0.09 | 24.31 | 31.55 | 6.33 |
| RN101 | Test acc↑ | 71.57 | 54.49 | 74.85 | 59.62 | 11.67 |
| | Unlearn acc | 68.20 | 56.00 | 97.00 | 85.40 | 18.00 |
| | Unlearn score↓ | 3.37 | 1.51 | 22.15 | 25.78 | 6.33 |

Table 21: Performance evaluation on sample unlearning on Mini-Imagenet dataset.

Table 21 shows the results of sample unlearning on Mini-Imagenet dataset. We did not report results of LCODEC because it requires excessive computation time. Goal of machine unlearning is to

remove influence of unlearning samples efficiently than re-training the model. However, LCODEC on Mini-imagenet requires at least two times of computational time than re-training the model.

Contrastive unlearning shows the low unlearn score, meaning it successfully altered embeddings of unlearning samples similar to test samples. Finetune is ineffective as it failed to reduce unlearn accuracy similar to the test accuracy. Gradient ascent has significant reduction in the test accuracy. Overall, contrastive unlearning is the only unlearning method that was able to properly reduce influence of unlearning samples.

## D.10 HYPERPARAMETER STUDY

We explore how batch size ($B$) and $\omega$ affect contrastive unlearning. Figure 3 and 4 show accuracy on test set (test accuracy, solid line) and test accuracy on unlearning samples (unlearn accuracy, dotted line) of random sample unlearning on CIFAR-10 dataset. Dots in each plot indicate where the algorithm determined its stopping point. As each figure shows, running the unlearning algorithm beyond the stopping point is not desired because it decreases model performance (low test accuracy), and unlearning samples show very different behavior than test data (bad unlearning effectiveness). The figures show that batch size heavily affects the performance of unlearning. This aligns with Graf et al. (2021). Contrastive unlearning loss is a batched process, and directions to pull and push are chosen based on the samples in the batch.

Figure 3 shows effects of different $\omega$ on unlearning process. $\omega$ is a hyperparameter that determines the number of contrasts for each batch of unlearning samples against batches of retain samples. Higher $\omega$ means each batch of unlearning samples is contrasted with many batches of retain samples. Higher $\omega$ stabilizes the unlearning procedure, however, which is computationally inefficient. All figures in figure 3 shows the algorithm achieves higher performance with a higher $\omega$. This shows higher $\omega$ stabilizes the unlearning process by reducing bias.

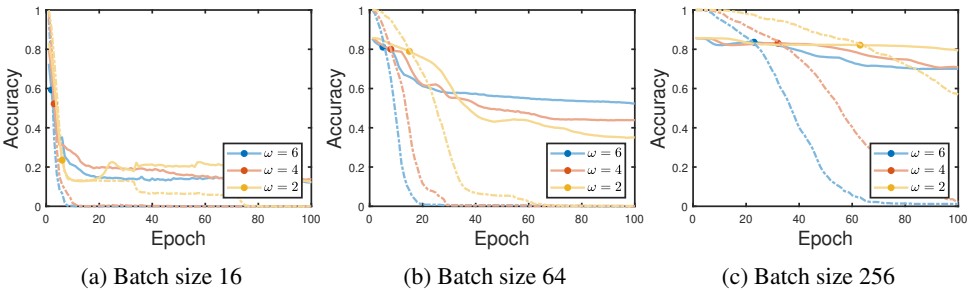

|              |              |              |
|:------------:|:------------:|:------------:|
| (a) Batch size 16 | (b) Batch size 64 | (c) Batch size 256 |

Figure 3: Test accuracy (solid line) and unlearn accuracy (dotted line) of contrastive unlearning on CIFAR-10 dataset from ResNet18. Each figure plots experiments on fixed batch size with different $\omega$.

Figure 4 shows the effects of different batch sizes on the unlearning process. A larger batch offers better stabilization as it reduces bias. When batch size is small, each unlearning sample in a batch is contrasted only with a small number of retain samples. On the other hand, if the batch size is larger, each unlearning sample is contrasted with more retain samples; hence, the directions to pull and push are less biased by retain samples. This leads to better model performance. However, a bigger batch is not always better as it requires more computation. Figure 4a, 4b, and 4c show that a batch size of 256 needs three times more iterations than a batch size of 64, while the test accuracy of two models from each plot is not much different.

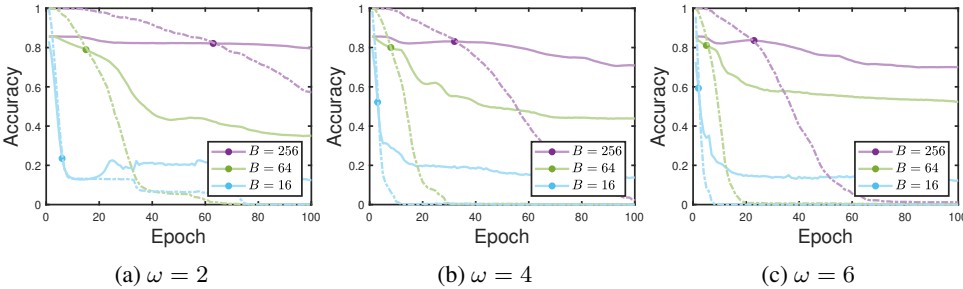

(a) $\omega = 2$       (b) $\omega = 4$       (c) $\omega = 6$

Figure 4: Test accuracy (solid line) and unlearn accuracy (dotted line) of contrastive unlearning on CIFAR-10 dataset from ResNet18. Each figure plots experiments on a fixed $\omega$ and different batch sizes.

