# OpenReview forum: "Contrastive Unlearning: A Contrastive Approach to Machine Unlearning"
_ICLR.cc/2025/Conference — ICLR 2025 Conference Withdrawn Submission_

### Official Review · Reviewer_i83Z · 2024-10-31

**Soundness:** 3
**Presentation:** 2
**Contribution:** 3
**Rating:** 5
**Confidence:** 1

**Summary:**

In this paper, the authors design a contrastive unlearning method. This method could remove the influence of unlearning samples by constrastive their embeddings against the remaining samples' embeddings.

**Strengths:**

1. The paper is well-organized.

2. The motivation is clearly described.

3. The appendix is comprehensive.

**Weaknesses:**

1. Table 1 is confusing. The results for “remain test” on RN18 seem to suggest that higher is better, so why is 85.79 bolded?

2. A similar issue appears in Table 3.

3. The datasets used are only CIFAR-10 and SVHN. It is recommended to include additional datasets.

4. The authors should release the code to ensure reproducibility.

**Questions:**

Please see weakness.

---

> ### Author Response · Authors · 2024-11-22
>
> We are very grateful to the reviewers for the detailed and insightful comments. We provide detailed point-to-point responses under each review.
>
> **Weakness 1 & 2**
> For Table 1 and 3, the retrained model is not a baseline, but a reference to unlearning, which provides the gold standard. Our goal is to compare all the unlearning algorithms including our contrastive unlearning and the other four baselines using the retrained model as a reference. 85.79 is highlighted because it’s the best performing among all the unlearning algorithms (excluding the reference results by the retrained model).
>
> **Weakness 3**
>  We also have conducted experiments on the Mini-Imagenet dataset, which are deferred to Appendix due to space limit. Please see Appendix D.7.
>
> **Weakness 4**
> We have provided the link to the code in Appendix C. Here is the [link to the implementation](https://anonymous.4open.science/r/contrastive_unlearning-22BF/README.md).

---

> ### Comment · Reviewer_i83Z · 2024-11-27
> **Official Comment by Reviewer i83Z**
>
> After reading the author’s response and the feedback from other reviewers, I’ve decided to keep my score unchanged. Additionally, as I’m not very familiar with this field, I hope the AC will give more weight to the opinions of other reviewers.

---

### Official Review · Reviewer_hNfr · 2024-11-01

**Soundness:** 2
**Presentation:** 1
**Contribution:** 2
**Rating:** 5
**Confidence:** 2

**Summary:**

The author introduces a novel form of contrastive loss tailored to the machine unlearning paradigm. The main insight is to selectively remove the most relevant features of the unlearning samples while preserving the quality of embeddings for the remaining samples. Experimental results demonstrate that the proposed contrastive loss achieves superior performance on single-class and sample-unlearning tasks, surpassing state-of-the-art methods.

**Strengths:**

1. The proposed supervised contrastive learning framework is well-structured and easy to understand. By constructing positive and negative pairs, the encoder effectively deactivates the embedding of unlearning samples.
2. The framework performs well across three benchmarks, showing improved unlearning efficiency and effectiveness over existing methods.

**Weaknesses:**

1. The approach of using supervised contrastive learning for unlearning is somewhat simplistic, as it overlooks the potential generalization decay of the model, as noted in Question 1. Merely introducing a framework that achieves improved performance is not sufficiently insightful. ICLR standards are high, and this work could benefit from a deeper exploration of these issues for the community.
2. The presentation quality could be enhanced, especially regarding punctuation usage. For example, there is a missing comma after Eq. (5) and an extra period before Eq. (6).

**Questions:**

1. Is the metric in Eq. (1) for machine unlearning reasonable? In the case of a well-performing pre-trained model, strong generalization should allow it to handle unseen samples effectively without degradation.
2. In Tables 2 and 5, the efficacy metric is the retraining time. However, retraining time can vary due to multiple factors, such as I/O rates and the number of GPU cores, which may affect the reliability of this metric.
3. Given that cross-entropy loss is included in the retraining, why is the processing time less than other baselines?

---

> ### Author Response · Authors · 2024-11-22
>
> We are very grateful to the reviewers for the detailed and insightful comments. We provide detailed point-to-point responses under each review.
>
> **Question 1**
> Yes. As retrained models (models that have been trained only on retain dataset) are gold standards of machine unlearning, we aimed to provide the problem definition accordingly.
>
> * Single class unlearning:  A retrained model without any sample of the unlearning class will never makes any prediction to the unlearning class even when it is given a sample of the class. Thus we defined the problem of single class unlearning with Eq. 1 and Eq. 2.
>
> * Sample unlearning: For a retrained model (without seeing any unlearning samples), the unlearning samples are essentially the same as the test samples that the model has not seen. Hence  we defined the problem of sample unlearning with Eq. 3. and Eq. 4.
>
> Technically, strong generalization should allow a model to handle unseen samples effectively without degradation.  In this case, sample unlearning will be easier to achieve.
> Numerous studies on membership inference attack also support that generalization reduces memorization of specific samples of the model.
>
> **Question 2**
> We are aware that the wall time is due to multiple factors and we used the same hardware and setup for our approach and all baselines to ensure a fair comparison. While there are alternative metrics that can be used for comparing the processed time (FLOPs, number of iterations over the dataset), they are not appropriate for the following reasons.
>
> * Each algorithm has unique computation that cannot be directly compared via other metrics. For example, UNSIR requires a noise calibration stage before iterating over the unlearning samples. Also, LCODEC requires hessian approximation and matrix inversion.
>
> * Most of the baselines did not address the stopping condition. For a fair comparison, we applied our stopping condition to baselines. Each of these baselines needs a different number of iterations over the unlearning set. For example, our contrastive unlearning requires 10 to 15 iterations over the unlearning dataset for unlearning 500 samples, while fine tuning requires 40 to 50 iterations. Hence comparing the number of iterations is not informative of the efficiency.
>
> **Question 3**
> While other baselines also have used cross-entropy loss, they require more processing time because they optimize the cross-entropy loss on the entire retaining dataset, while we only minimize the cross-entropy loss on retain samples that are selected within the batch. Also they needed more iterations to achieve the stopping condition.

---

> > ### Comment · Reviewer_hNfr · 2024-11-27
> >
> > After reading other reviews and the response, I still believe that the paper falls slightly below the acceptance threshold.  I am inclined to keep my score.

---

### Official Review · Reviewer_9CZd · 2024-11-02

**Soundness:** 3
**Presentation:** 3
**Contribution:** 3
**Rating:** 5
**Confidence:** 2

**Summary:**

The paper introduces a novel framework for machine unlearning, which is the process of removing the influence of specific training samples from a trained model. The authors propose a method called contrastive unlearning that leverages representation learning to effectively eliminate the impact of unlearning samples while maintaining the model's performance on the remaining data. The approach contrasts the embeddings of unlearning samples with those of the remaining samples, adjusting the model's representation space to make the unlearning samples' embeddings similar to those of unseen test samples. The paper concludes that contrastive unlearning is a promising technique for machine unlearning, with potential applications in complying with privacy regulations. Through extensive experiments on various datasets and models, the paper demonstrates that contrastive unlearning outperforms existing methods in terms of unlearning efficacy, model performance preservation, and computational efficiency.

**Strengths:**

1. The paper introduces contrastive learning to the machine unlearning, effectively removing the influence of unlearning samples without significant loss in model performance. Contrastive unlearning is computationally efficient, requiring fewer iterations to achieve the desired unlearning effect compared to other methods.

2. The experiments are comprehensive, including accuracy gap and membership inference attack for unlearning performance. They convinced me of the SOTA performance of the proposed contrastive paradigm.

**Weaknesses:**

1. Weak scalability compared to non-contrastive methods. The proposed contrastive learning method constructs the positive pair and negative pair in a batch, hence, it might need a large batch size in the practical application which has a large class number. So I suggest using a MoCo-like, more advanced contrasive learning method to further enhance the scalability of the proposed method.

2. The assumption for test samples is a little bit strong, e.g., " If the embeddings of the unlearning samples become indistinguishable from the embeddings of the test samples, we can claim that the model is no longer influenced by the unlearning samples". We usually assume the training and test examples follow I.I.D. However, it seems that this paper did not have this assumption.

**Questions:**

1, Could you provide some discussion for the limits regarding the number of unlearning samples. More unlarning samples in the contrastive framework might damage the representation of ramaining samples.
2. Following the weakness 2, do the existing methods have the same assumption as line 62-65?

---

> ### Author Response · Authors · 2024-11-22
>
> We are very grateful to the reviewers for the detailed and insightful comments. We provide detailed point-to-point responses under each review.
>
> **Regarding weakness 1**
> It is true that the batch size affects the performance of our contrastive unlearning. As the reviewer had suggested, bigger batch size is often necessary for achieving better performance and unlearn efficacy. /We consider that contrastive unlearning is not limited to a particular contrastive learning algorithm, as our key idea is to push unlearning samples away from their own classes. We deem that most contrastive learning algorithms can be used. To show this, we slightly modified MoCo for contrastive unlearning and conducted sample unlearning (500 samples) from ResNet18. Here are the results. [Please refer to the link for the implementation](
> https://anonymous.4open.science/r/contrastive_unlearning_moco-357D/).
>
> | Test accuracy | Unlearn accuracy |
> |--------------------|-------------------------|
> |     76.04          |           71.8            |
>
> **Weakness 2**
> We assumed that training samples and test samples follow I.I.D. We did not assume the test samples to have unique distributions. We will clarify that in our revision.
>
> **Question 1**
> In the appendix, we provide experiments on different numbers of unlearning samples. When the number of unlearning samples is large, model performance reduces because more remaining samples are adversely affected. We have observed that contrastive unlearning is capable of achieving good unlearn efficacy (Eq. 3 from the section 3.2) with the cost of model performance. Thus we deem that the magnitude of performance loss would determine the limit of number of samples. Please refer to Appendix D.4 for further discussion.
>
> **Question 2**
> Some of the recent unlearning studies have been discussing embedding space and decision boundaries for their analysis. For example, [Cha et. al (2024)] also argued that unlearning samples have different distribution with test samples, and it is important to push them to the boundaries to achieve unlearning effect. However, unlike our analysis, the authors defined the goal of unlearning as incorrectly classifying all unlearning samples. While we consider that unlearning samples should be seen as unseen samples (test samples) to the model.
>
>
>
> Sungmin Cha, Sungjun Cho, Dasol Hwang, Honglak Lee, Taesup Moon, and Moontae Lee. Learning to unlearn: Instance-wise unlearning for pre-trained classifiers. Proceedings of the AAAI Conference on Artificial Intelligence, 38(10):11186–11194, Mar. 2024. doi: 10.1609/aaai.v38i10.28996. URL https://ojs.aaai.org/index.php/AAAI/article/view/28996.

---

> ### Comment · Reviewer_9CZd · 2024-11-28
> **Feedback of Author Response**
>
> Dear authors:
>          Thanks for your efforts to clarify my concerns and questions. After reading the other comments and your feedback, I still feel that the work can have a better version regarding the discussion of scalability and the assumption. So I tend to reduce my score from 6 to 5.

---

> ### Author Response · Authors · 2024-11-30
>
> Thank you for your comment. We would like to further clarify regarding the scalability and the assumption.
>
>
> ### Scalability
> * Unlike standard contrastive learning, contrastive unlearning does not necessarily require a large batch size or a large negative sample size.  Through the ablation study, we experimented with batch sizes of 64, 128, and 256 (Appendix D.10). Test accuracy significantly increased when the batch size increased from 64 to 128, but did not increase much from 128 to 256. Our observations on batch size and utility preservation were consistent across datasets, including CIFAR-10, SVHN, and even with the Mini-ImageNet which has 100 classes. A batch size of 128 is not particularly large and does not pose significant scalability concerns. It is a standard choice frequently employed in various experiments and practical applications. In addition, contrastive unlearning requires significantly fewer iterations.  Hence the contrastive unlearning approach is much more scalable and effective compared to non-contrastive approaches as we have demonstrated in the experimental results.
>
> *  For example, in Table 2, we used a batch size of 128 for Contrastive, Boundary Shrink and Boundary Expansion, 256 for Scrub and Unsir, through a grid search for best performance. Contrastive unlearning needed only one full iteration over the entire unlearning samples whereas Boundary Shrink, Boundary Expansion and Scrub needed one to three iterations. For Unsir, although it only requires one iteration, it also needs a noise calibration stage. Similarly, in Table 5, we used a batch size of 128 for Contrastive and Gradient Ascent, 256 for finetune and LCODEC. Fisher conducts a one-time hessian gradient update for entire unlearning samples to achieve unlearning. However, obtaining hessians cannot be accelerated via batch processing and must be done by iterating each sample. Contrastive typically satisfied the stopping condition with 10~15 iterations whereas Finetune needed at least 40 iterations. Gradient Ascent needs 10 to 40 iterations. While LCODEC only requires 10 iterations, each iteration is computationally expensive. Both tables demonstrate that contrastive unlearning is significantly more efficient than baselines.
>
>
> * Our contrastive unlearning framework is general and can be implemented with any contrastive algorithms as we have demonstrated. Hence even when a large number of negative pairs is needed, MoCo-based implementations can be utilized to further scale up the performance.
>
>
> ### Assumption
>
> Regarding the assumption, as mentioned, we do not have any specific assumptions about the test data.  “If the embeddings of the unlearning samples become indistinguishable from the embeddings of the test samples, we can claim that the model is no longer influenced by the unlearning samples".  This is not an assumption, instead it’s our unlearning goal.  In the ideal case, if a model is retrained without the unlearning sample, the unlearning sample would behave like an unseen test sample.  Our goal is to remove the specific memorization of the unlearning samples by the model.

---

### Official Review · Reviewer_Hf8Q · 2024-11-06

**Soundness:** 3
**Presentation:** 2
**Contribution:** 3
**Rating:** 5
**Confidence:** 2

**Summary:**

In this paper, the authors propose a new machine unlearning method, inspired by contrastive learning.

The proposed contrastive unlearning pushes away the positive pairs (from the same class) and pulls the negative pairs close to each other.

The idea is validated on several backbones and datasets.

**Strengths:**

- It seems that the contrastive paradigm is new for the field of unlearning.
- The experimental results, from both efficiency and efficacy, seem to show the advantages of the proposed contrastive unlearning (while I'm quite confused about *unlearning acc*, which will be elaborated in Weaknesses and Questions).

**Weaknesses:**

I'm not familiar with machine unlearning. I read some papers from recent ML conferences, such as ICML, NeurIPS, and CVPR, and try my best to provide a fair review. Some comments may be naive and I'd like to update my score after reading the response from the authors and reviews from other reviewers. Here are my concerns:

- I'm quite confused about the setting of single class unlearning. If we expect a model to forget a class, why don't we just add some rules? For a simple classification task, the rule could be very simple: assign random labels if the model outputs the class to unlearn. The rule may be useless for more powerful models, such as zero-shot classifiers (e.g., CLIP) and LLMs. However, the authors fail to conduct experiments on these models.

- Some visualization may be helpful for the "contrastive" unlearning. It is quite common to show some t-SNE visualization in contrastive learning. With the visualization, we can know whether the unlearning samples are pushed to the decision boundary.

- In Eq. (5) and Eq. (6), $z_i$ is not normalized, which is a standard step in contrastive learning. In CL, $z_i$ is usually processed by $z_i = \frac{z_i}{\|\|z_i\|\|}$. Is it a special setting to ensure the performance?

- The theoretical analysis is lacking. Although it is not necessary, some theoretical analysis may help to improve the quality of this paper.

- There are some typos, such as  $ \mathcal D_{ts} \cup \mathcal{D}_{tr} = \emptyset $ in Line-198.

**Questions:**

- Why do the authors test the single class unlearning only on simple classification models?
- How will the embeddings of unlearning samples change? Are they really pushed to the decision boundary?
- If we normalize the representation $z_i$ in Eq. (5) and (6), how will the performance change?

---

> ### Author Response · Authors · 2024-11-22
>
> We are very grateful to the reviewers for the detailed and insightful comments. We provide  detailed point-to-point responses under each review.
>
>
> **Weakness 1 & Question 1**
>  A rule-based approach to class unlearning by randomly reclassifying any sample of the specified class, might be plausible in certain scenarios, such as Machine Learning as a Service (MLaaS), where only the classification result is accessible. However, it can only achieve superficial unlearning in the more commonly considered scenarios in the machine unlearning literature, where model weights are accessible and must therefore be scrubbed. Specifically, the rule-based approach has significant limitations and flaws under these prevalent machine unlearning settings:
>
> * Insufficient unlearning: Privacy regulations mandate that the influence of the data be completely erased. Simply reclassifying samples does not remove the influence of the specified class from the underlying model weights.  Patterns, correlations, or features learned from the removed class remain embedded in the model. If the model is leaked, an adversary can still recover samples associated with the class in the training data.
>
> * Model utility: If a model is retrained without the specified class C, the decision boundary might change for all remaining classes. For example, samples of class A which previously had a high probability of being misclassified as C may have a high probability to be classified as A in the retrained model.  Unlearning aims to achieve similar performance as the retrained model.  In contrast, random reclassification would still misclassify samples of A to class C, then reclassify them into a random label, hence potentially reducing the performance of all remaining classes.
>
> For these reasons, our study, along with  other unlearning studies, aim to modify model weights to achieve the effect of class unlearning. We will add discussions on rule-based unlearning and point out its potential in certain scenarios such as MLaaS. Thank you for your suggestion.
>
> **Weakness 2**
> We attempted visualization using t-SNE. However, there are several reasons that made it infeasible:
>
> * To show the progress of unlearning, we attempted to obtain multiple t-SNE plots for different timestamps of unlearning. Due to the non-convex nature of t-SNE algorithm, plot locations (location of each class) are not fixed for each plot. Thus samples of each class appeared in different locations for every plot.
>
> * t-SNE algorithm effectively provides a global view of how well all classes are separated from each other. However, it is not effective for showing the sample level distributions.
>
> * Distance between clusters of t-SNE plots does not represent the actual distance. Our contrastive unlearning mechanistically pushes the unlearning samples away from their class distribution. In order to effectively visualize this, it is crucial to visualize that the distance between unlearning samples and their classes is increasing. However, the distance between points in t-SNE plot does not depict the distance between the two samples.
>
> For these reasons, we consider that including t-SNE plots would cause confusion instead of clarification. Thus we decided not to include the figure.
>
> **Weakness 3 & Question 3**
> Similar to contrastive learning algorithms, we normalize z_i in embedding space before computing the unlearning loss. Normalizing z_i is crucial for manipulating the embeddings and it is the general requirement for contrastive learning algorithms. If z_i is not normalized, then unlearn efficacy would decrease significantly.
>
> **Weakness 4**
> Thank you for pointing them out. We will fix the typos.
>
> **Question 2**
> Yes. They move towards the decision boundary. From sample unlearning results (Table 1), accuracy on unlearning samples substantiates that they have moved towards the decision boundary.
>
> **Additional experiments for weakness 1**.
> We conducted additional experiments to see whether contrastive unlearning can remove a class of CIFAR-100 dataset from more powerful models like few-shot CLIP. We first finetune the pre-trained CLIP with CIFAR-100 to achieve top-1 accuracy of 81.2%. Then we unlearn class 1 using contrastive unlearning. Results show that our contrastive unlearning is capable of unlearning the class, validating the generalizability of the approach. Please refer to the [following link for the code](https://anonymous.4open.science/r/few_shot_contrastive_unlearning-ICLR2025/readme.md)
>
> Test accuracy of fine tuned CLIP
>
> | Test accuracy | Unlearn accuracy |
> |--------------------|-------------------------|
> |        81.2         |       78.3                |
>
> Test accuracy and unlearn accuracy (accuracy on unlearning data (class 1))
>
> | Test accuracy | Unlearn accuracy |
> |--------------------|-------------------------|
> |         65.0           |           0.0              |

---

> > ### Comment · Reviewer_Hf8Q · 2024-11-27
> >
> > I thank the authors for the response. After reading it and other reviews, I believe that the paper needs a significant revision to clarify the weaknesses mentioned above, though the response partially addressed some of my concerns. I keep my score unless other reviewers provide more persuasive comments to recommend acceptance.

---

### Note · Authors · 2025-01-05

**Comment:**

Dear reviewers and PC chairs,

After a discussion, we have decided to withdraw our paper from the current ICLR 2025 submission.
We sincerely appreciate the time and effort you have devoted to reviewing our work.

Best,

**Withdrawal Confirmation:**

I have read and agree with the venue's withdrawal policy on behalf of myself and my co-authors.